# Temperature-adaptive hydrogel optical waveguide with soft tissue-affinity for thermal regulated interventional photomedicine

Guoyin Chen[1], Kai Hou [1] ✉, Nuo Yu[1], Peiling Wei[1], Tao Chen[1], Caihong Zhang[1], Shun Wang[1], Hongmei Liu[1], Ran Cao [1] ✉, Liping Zhu[1], Benjamin S. Hsiao [1,2] & Meifang Zhu [1] ✉

Photomedicine has gained great attention due to its nontoxicity, good selectivity and small trauma. However, owing to the limited penetration of light and difficult monitoring of the photo-media therapies, it is challenging to apply photomedical treatment in deep tissue as they may damage normal tissues. Herein, a thermal regulated interventional photomedicine based on a temperature-adaptive hydrogel fiber-based optical waveguide (THFOW) is proposed, capable of eliminating deeply seated tumor cells while lowering risks of overtemperature (causes the death of healthy cells around the tumor). The THFOW is fabricated by an integrated homogeneous-dynamic-crosslinking-spinning method, and shows a remarkable soft tissue-affinity (low cytotoxicity, swelling stability, and soft tissue-like Young's modulus). Moreover, the THFOW shows an excellent light propagation property with different wavenumbers (especially −0.32 dB cm$^{-1}$ with 915 nm laser light), and temperature-gated light propagation effect. The THFOW and relevant therapeutic strategy offer a promising application for intelligent photomedicine in deep issue.

In recent years, as an emerging technology, photomedicine has been used for treating cancerous diseases with advantages of small trauma, nontoxicity, and good selectivity[1,2]. The treatment is principally based on the light-induced physical reactions (generation of heat, such as photothermal therapy)[3,4], chemical reaction (photochemical reaction)[5], or biological processes (optogenetic, photobiological)[6,7] onto the disease location by inducing exogenous photosensitive reagent[3,8,9]. However, as the light penetrates only a few centimeters through the tissue, it is difficult to apply photomedical treatment in deep tissue[10–12].

Interventional therapy is one of the most effective clinical treatments to replace the deeply invasive surgical operations[13], which relies mainly on an intervened medium (metal wire, silica optical fiber, etc.) to import apparatus into the deeply seated disease location. And in vitro equipment (such like computed tomography angiography, ultrasound, magnetic resonance, and etc.) is combined to realize diagnosis and treatment[14,15]. In consequence, in the way that light-guides used for interventional therapy, photomedicine could effectively solve the problem of light penetration through tissue. Although light-guides such as silica- and polymer-based optical fibers have been proven to be feasible, their stiffness and poor biocompatibility may cause inflammation or damage to host tissue[12,16]. Besides, during photomedical treatment, the photo-induced effects should be controlled in a manner to avoid the damage of normal tissue around the diseases location[9,17]. To this end, combining equipment assisted imaging technique with interventional photomedical treatment to monitor and guide the therapeutic process is an efficient way[2,14]. However, this kind of treatment is high-cost, time-consuming and complicated.

[1]State Key Laboratory for Modification of Chemical Fibers and Polymer Materials, College of Materials Science and Engineering, Donghua University, 2999 North Renmin Road, Shanghai 201620, China. [2]Department of Chemistry, Stony Brook University, Stony Brook, NewYork, NY 11794, USA. ✉e-mail: houkai711@dhu.edu.cn; rancao@dhu.edu.cn; zmf@dhu.edu.cn

Hence, an efficient method for dynamic and precise in situ physiological microenvironmental monitoring (temperature, pH, etc.) is desired for guiding the photomedical therapeutic process in a controlled manner under the deep tissue.

Hydrogel-based optical Waveguide are ideal candidates for using as intervened light-guides for accurate and targeted light propagation[6,18,19]. As a matrix, hydrogels can be extensively modified by molecular design and realize environmental responses, such as temperature, pH, and molecular response[20,21]. In this regard, fabricating hydrogel optical waveguide with a certain condensed structure and adjustable optical properties, could realize light guiding and physiological microenvironmental monitoring simultaneously. For example, based on the "coil to globule" induced transparency-opacity transition at the lower critical solution temperature (LCST), thermosensitive hydrogels could be used as temperature-dependent optical switches[21]. However, monomers were not preferred for fiber forming due to insufficient interaction. Furthermore, inevitable phase separation during exothermic free radical polymerization would induce agglomeration of molecular chains or cross-linked microregions, which are disadvantageous for functional hydrogel fiber formation, thus it is difficult to obtain a hydrogel fiber with a uniform structure.

In this work, a desired temperature-adaptive hydrogel fiber-based optical waveguide (THFOW) is fabricated by an integrated homogeneous-dynamic-crosslinking-spinning on large-scale. The fabricated THFOW shows an excellent light propagation property with different wavenumbers (especially −0.32 dB cm$^{-1}$ of light attenuation with 915 nm laser light) and highly sensitive temperature-gated light propagation effect. In addition, a thermal regulated interventional photomedicine based on the THFOW is demonstrated, capable of eliminating deeply seated tumor cells while lowering the risks of overtemperature. In consequence, the fabricated THFOW shows a great potential for application in the field of intelligent photomedicine.

## Results and discussion
### Concept of thermal regulated interventional photomedicine

A concept of thermal regulated interventional photomedicine (Fig. 1) is proposed here, which is capable of efficiently eliminating the tumor cells while lowering the risks of the overtemperature that causes the death of normal cells around the tumor site. First, with remarkable soft tissue-affinity, the fabricated THFOW can be implanted into deep tissues of human body and target the tumor site without inflammation. The uniform gel structure and high transparence enable the THFOW efficiently transport 915 nm laser light from external laser equipment to the disease site in vivo. Consequently, photothermal heating around the tumor is induced as pre-injected photothermal nanoparticles

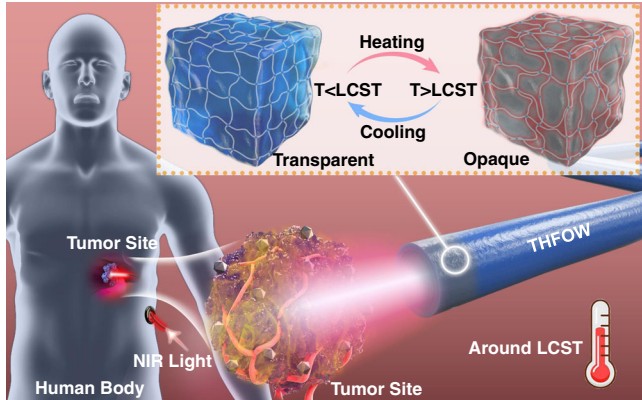

**Fig. 1 | A concept of thermal regulated interventional photomedicine.** Schematic illustration of the thermal regulated interventional photomedicine (LCST means the lower critical solution temperature) in deep tissue by using the temperature-adaptive hydrogel fiber-based optical waveguide (THFOW).

exposing to 915 nm NIR laser. When the temperature of the tumor site approaches LCST of the THFOW, phase separation would occur between the polymer network and H$_2$O within the core hydrogel fiber, causing a reduced transparency of THFOW (Fig. 1 top-right part), and the 915 nm NIR laser would scatter the surrounding tissue. Hence, the temperature of the tumor site can be regulated around the lower critical solution temperature (LCST) of the THFOW. At this temperature, the cancer cells will be efficiently killed, while avoiding serious damage to the surrounding normal tissue caused by overtemperature. Thus, the fabricated THFOW showed great application potential in the field of intelligent photomedicine.

### Continuous synthesis and structural characterization of the THFOW

The key of the thermal regulated interventional photomedicine in deep tissue is fabricating an optical hydrogel fiber with low light attenuation and tunable thermosensitivity. Here, we chose N-isopropylacrylamide (NIPAM) as the functional monomer, and N,N-dimethylacrylamide (DMAAm) as the hydrophilic component to tune the LCST of the fabricated hydrogel. To obtain a low light guiding attenuation, the optical transparency of the raw materials is an important parameter, and the core should have higher refractive index than the sheath[6,18]. All the NIPAM/DMAAm (ND) hydrogels intuitively showed a high transparency from the photographs of the $(N_XD_{100-X})_{50}$ hydrogels (Supplementary Fig. 1a), and Supplementary Fig. 1b shows the transmittance and photographs of the $(N_XD_{100-X})_{50}$ hydrogels. All the hydrogels showed a high transmittance (>90%) in the 460–920 nm range, indicating high transparency, and thus guaranteed low light attenuation[6,11,18]. In addition, the LCST of the different $(N_XD_{100-X})_{50}$ hydrogels was investigated to select a suitable monomer composition for the pre-gel solution (Supplementary Fig. 2); when the temperature was higher than the LCST, phase-separation-caused opacity was observed in $(N_XD_{100-X})_{50}$ hydrogels (Supplementary Fig. 2c), which showed a sensitive temperature-controlled switch for light transmittance. Furthermore, it can be seen that with the increase in DMAAm, the LCST of the fabricated hydrogel increased, and coincided with the formula of LCST = 31.37 + 0.58 × $C_D$, where $C_D$ is the concentration of DMAAm in total monomers, caused by increase in DMAAm (hydrophilicity) content. As the hydrophilic component in the polymer network increased, it requires a higher temperature to trigger the phase-separation[21,22]. Considering the suitability of transmittance and thermosensitivity, and 48 °C was the temperature that could efficiently eliminate the cancer cells, thus we chose $(N_{70}D_{30})_{50}$ (LCST = 48.6 °C) as the precursor composition to fabricate the desired THFOW.

THFOW was fabricated using an integrated homogeneous-dynamic-crosslinking-spinning method. As illustrated in Fig. 2a, 2 wt% Na-alginate solution and $(N_{70}D_{30})_{50}$ solution were used as sheath and core spinning solution, respectively. The forming processes mainly consist of two stages: during the first stage, the Na-alginate solution gelled immediately into the Ca-alginate sheath hydrogel fiber when it was injected into a CaCl$_2$ coagulating bath (~ 10 °C, as illustrated in the bottom-right part of Fig. 2a)[20,23]. During the second stage, the sheath hydrogel tube brings the core monomer solution into the UV light region, which triggers radical polymerization of double bonds among the monomers (NIPAm, DMAAm, and PEGDA), forming the core hydrogel polymer networks (as illustrated in the top-left part of Fig. 2a). However, owing to the heat released from the polymerization, the temperature would increase sharply over the LCST of the pre-gelled NIPAM-based hydrogel without timely heat dissipation, causing phase separation within the thermosensitive pre-gelled core monomer solution. Thus, polymerization occurs in two different phases, leading to inhomogeneous polymer networks[11]. To address this problem, the coagulating bath was cooled using ice during the spinning process (<10 °C), which could promote the dissipation of the released heat (Fig. 2b), thus avoiding phase separation during the polymerization

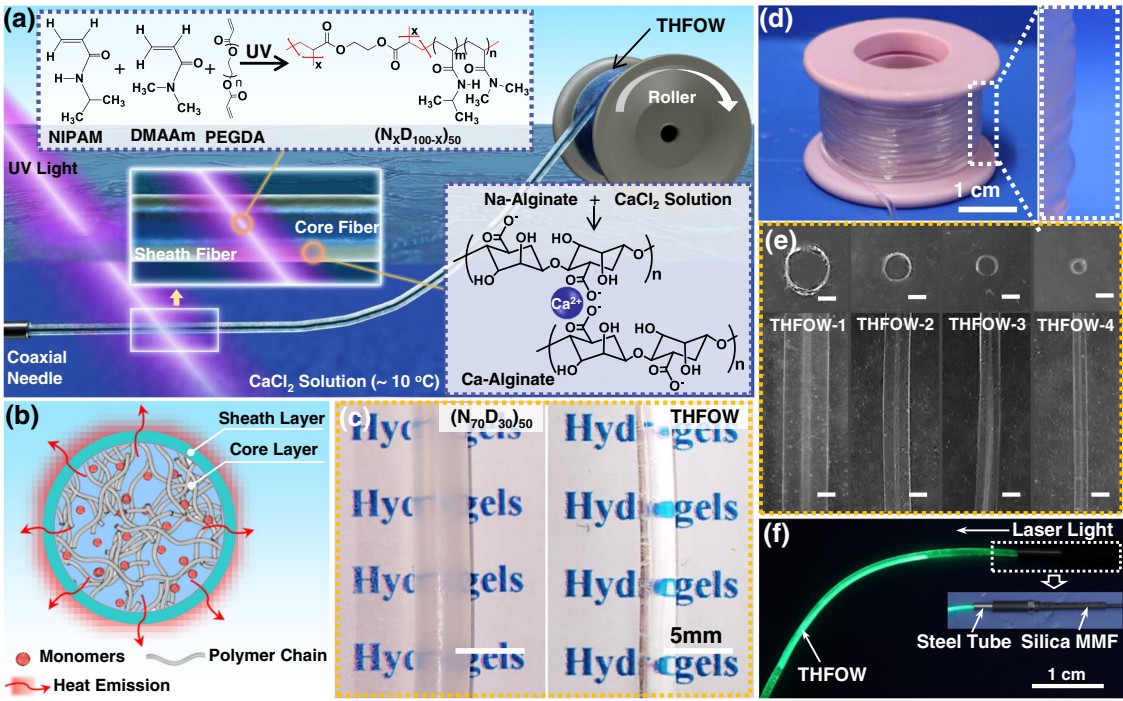

**Fig. 2 | Fabrication and characterization of the THFOW.** Schematic illustration of (**a**) the fabricating process of the THFOW and (**b**) heat emission from the core monomer precursor; (**c**) Photos of the $(N_{70}D_{30})_{50}$ hydrogel and the fabricated THFOW; Photos of (**d**) a bobbin of the fabricated THFOW and (**e**) Cross-sectional and side view of the THFOW in deionized water, scale bar = 1000 μm; (**f**) Light transmission within the THFOW.

process. This guarantees uniformity of the polymer network within the THFOW. As a comparison, the $(N_{70}D_{30})_{50}$ hydrogel synthesized by direct UV polymerization (without cooling process) showed a foggy opacity (Fig. 2c, left), while that with cooling process, the fabricated THFOW showed highly transparency (Fig. 2c, right). After collecting on a roller and going through an aftertreatment to remove the residual monomers and impurities, THFOW was finally obtained (Fig. 2d). Moreover, THFOW with different diameters can be fabricated by varying the spinning needle with different diameters (Supplementary Fig. 3) from this method (Fig. 2e, Supplementary Fig. 4), and diameters of 2417 μm/2533 μm, 1395 μm/1490 μm, 1097 μm/1223 μm, and 633 μm/ 781 μm (core diameter/sheath diameter) could be realized, and named as THFOW$_{2500}$, THFOW$_{1500}$, THFOW$_{1200}$, THFOW$_{800}$ as shown in Fig. 2e. These THFOWs with different diameters are suitable for various application scenarios. It's worth noting that the thickness of the sheath layer is in the range of 50–60 μm for all the samples, which is much smaller than the corresponding sheath thickness of the spinning needles (Supplementary Fig. 3). This was because the sheath layer hydrogel tends to shrink for rigidly limiting the swelling of the core hydrogel (discussed in detail below).

### Evaluation of light propagation properties

This THFOW showed apparently excellent light propagation properties (Fig. 2f), which is due to its clear and uniform core-sheath structure (Supplementary Figs. 5, 6), and also because the refractive index (RI) of the core hydrogel was higher than that of the sheath hydrogel (Supplementary Fig. 7). Consequently, a total reflection would occur at the interface between the core and sheath hydrogel, forming a light propagation path through the THFOW[6,11].

As for the detailed evaluating light propagation properties of the THFOW, laser light was focused on one tip of THFOW, and the scattered light intensity was measured throughout the fiber length (~10 cm). The results showed that laser light with different wavelength ($\lambda$ = 450, 515, and 650 nm) could efficiently be propagated through THFOW (Fig. 3a and b; THFOW was ~10 cm in length). Furthermore,

analysis of the light intensity profile of scattered light through the THFOW showed that the light loss of all the THFOW with different diameters was in the range of 0.17 dB cm$^{-1}$ to 0.41 dB cm$^{-1}$, as shown in Fig. 3c. The light loss through the THFOW was one of the lowest among the other similar optical hydrogel fibers, such as step-index hydrogel optical fiber with fiber diameter of 800/900 μm (0.32 dB cm$^{-1}$ with 492 nm laser light)[18], strain-sensing hydrogel optical fibers with fiber diameter of 750/1100 μm (0.45 dB cm$^{-1}$ with 532 nm laser light)[24], and alginate@PAAm hydrogel fiber with fiber diameter of 500 μm (0.25 dB cm$^{-1}$ with 472 nm laser light)[7]. It can be seen that THFOW coupling with laser light of a larger wavelength would have lower attenuation, as the light with a larger wavelength had better transmission through THFOW (Supplementary Fig. 1b). Furthermore, the results also show that diameter of the THFOWs slightly affect the light loss; the larger the diameter, the smaller the light loss, which was due to the longer ray propagation distances before reflection at the sheath/ core interface within THFOW[6,11]. Supplementary Fig. 8a shows laser light with different power could all propagate through the THFOW, and the light loss of the light profiles shows that light power didn't affect the attenuation of the light in THFOW (Supplementary Fig. 8b). In addition, The THFOW could be used as an implantable light-guide to transport NIR light (Supplementary Fig. 9), however is partially detected by the digital camera due to that the NIR light is a kind of invisible light. Thus, we tested the light attenuation with the 915 nm NIR light through the THFOW by the method of cutback technique, the result in Fig. 3d shows that the measured light loss of the THFOWs in the air were in the range of −0.32 dB cm$^{-1}$ to 0.39 dB cm$^{-1}$.

### Temperature-gated light propagation effect

Thermosensitivity is a key property of the THFOW, which endows the optical hydrogel fiber with intelligent responsiveness to environmental temperature, thus realizing controllable photothermal cancer therapy in deep tissue. As shown in Fig. 4a, the THFOW showed excellent light propagation when it was partly immersed in a water bath with a temperature of 48 °C. When the temperature rose to 52 °C,

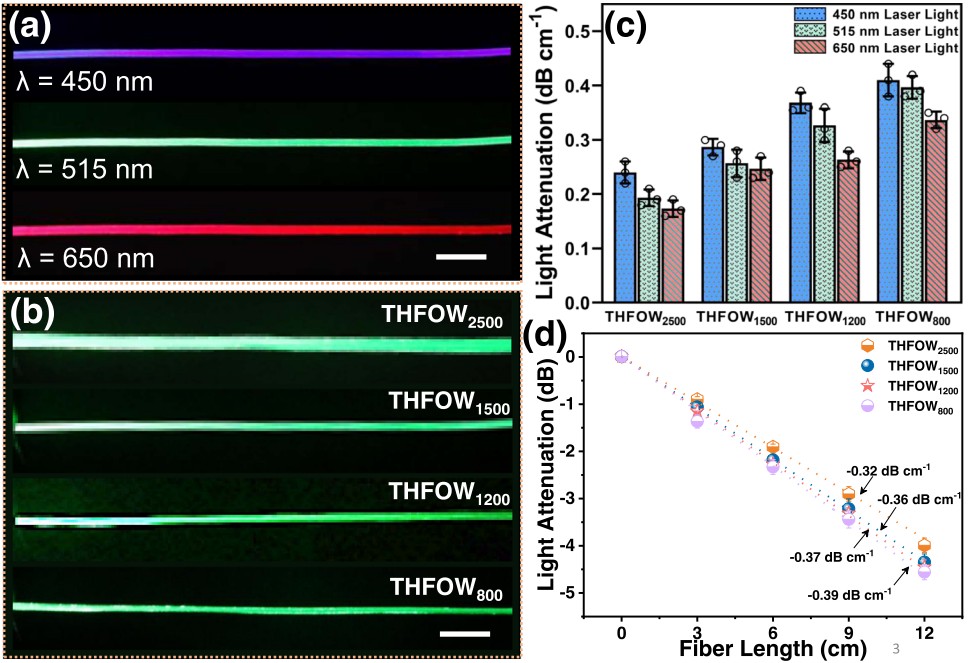

**Fig. 3 | Light propagation through the THFOW. a** Laser light with different wavelength ($\lambda$ = 450, 515 and 650 nm) propagate through the THFOW; (**b**) Laser light propagates through THFOWs with different diameters; (**c**) Light attenuation of the THFOW calculated from the scattered light intensity along with the THFOW profile ($n$ = 3 independent experiments), data were presented as mean ± SD; (**d**) Propagation loss of the 915 nm laser light through THFOWs, measured by a cutback technique ($n$ = 3 independent experiments), data were presented as mean ± SD. Scale bars in (**a**), (**b**) are 1 cm.

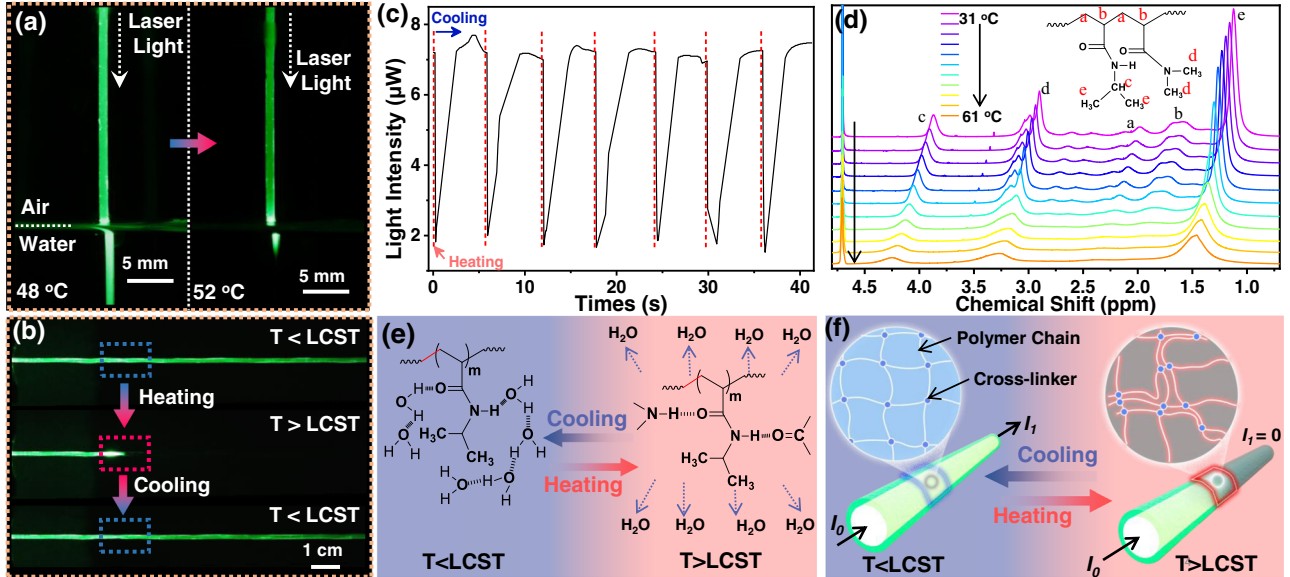

**Fig. 4 | Thermo-sensitivity of the THFOW. a** Photos of light propagation through THFOW in water under different temperatures (48 °C and 52 °C); (**b**) Photos of the light propagate through the THFOW with a segment of THFOW heating up and getting cool in sequence; (**c**) Light intensity variation during the cyclic heating and cooling of the THFOW segment; (**d**) Temperature-dependent $^1$H NMR spectra of ($N_{70}D_{30}$)$_{50}$ hydrogel in $D_2O$ from 31 to 61 °C with an increment of 3 °C; (**e**, **f**) Schematic illustration of the thermosensitivity of the THFOW.

the part immersed in water bath showed no light propagation, as LCST caused phase separation between the polymer network and surrounding water molecules. In other words, the opacity of the core hydrogel fiber drastically decreased, thus reduced the propagated light intensity (temperature-gated light propagation effect). To digitize the effect of phase-separation during light propagation, THFOW was coupled with 532 nm laser light and the light intensity was measured at the other tip (Fig. 4b, c). Initially, a light intensity of about

7 μW was detected. When a segment of it was heated, phase-separation-caused opaqueness significantly hindered the light propagation through THFOW (Supplementary Fig. 10), and only about 2 μW light intensity was detected; after cooling, photoconductivity could be recovered almost completely. Furthermore, this thermally induced light-switching process showed good cyclicity (Fig. 4c).

A temperature-variable $^1$H NMR analysis was used to further investigate the phase separation mechanism of the THFOW (Fig. 4d). It

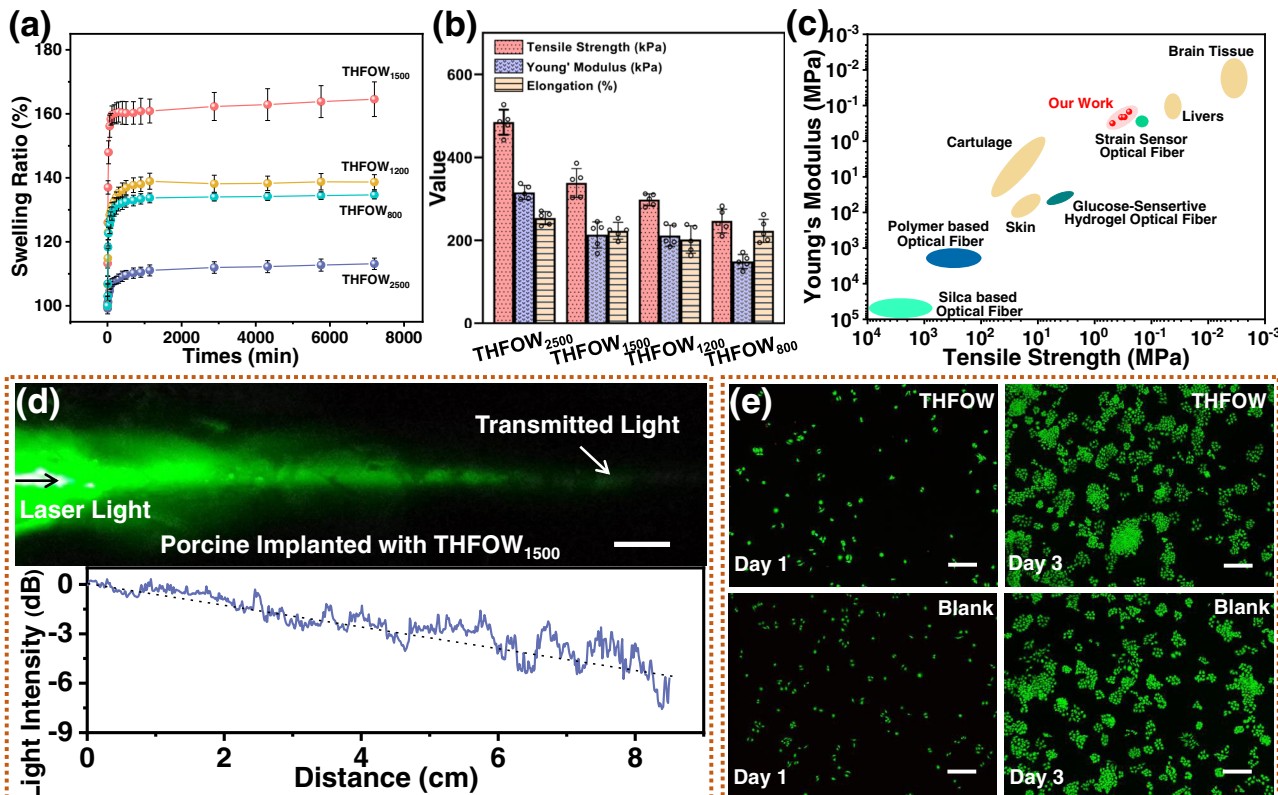

**Fig. 5 | Compatibility of the THFOW with organisms. a** Swelling behavior ($n = 3$ independent experiments) and (**b**) mechanical properties ($n = 5$ independent experiments) of the fabricated THFOWs, data in (**a**) and (**b**) were presented as mean ± SD; (**c**) Comparison of mechanical properties among the THFOW, tissues and other light-guide fibers; (**d**) Laser light ($\lambda = 515$ nm) propagation within porcine tissue through implanted THFOW$_{2500}$, scale bar = 1 cm. **e** Live/dead assay of Hela cells on THFOWs compared to blank at 1 and 3 days, where live cells are in green, Scale bars: 100 μm. A representative image of three independent samples from each group is shown in **e**.

can be seen that during the temperature-rise process (from 31 °C to 61 °C), the resonant peaks corresponding to the side chain ($H_c$, $H_d$, $H_e$) remained sharp, while the intensities of the peaks from the backbone ($H_a$, $H_b$) were severely reduced and almost disappeared. The reason can be explained that the side chain possesses relatively lower dehydration and higher hydrogen bonds than the backbone[25]. This indicates that there are different microenvironments between the main chain and the side chain. Furthermore, a phase-separated fraction ($p$) was used to quantitatively evaluate the phase separation in the THFOW[25], and the results are shown in Supplementary Fig. 11. It can be seen that when the temperature was higher than LCST, $p$ of the side chain would reach a much higher value than that of the main chain. This indicates that more water molecules surround the hydrophilic side chains, and fewer water molecules surround the backbone, as illustrated schematically in Fig. 4e. This difference in microenvironment within the THFOW (when the temperature > LCST) declined opacity of the core hydrogel fiber, and this hinders light propagation through THFOW. As a result, the laser light would scatter out of THFOW in the phase transition segments, cutting off the light propagation path through it. After the temperature falls below LCST, the transparency is recovered, hence, the light propagation path through the THFOW resumes (Fig. 4f). In order to show whether the temperature of the THFOW after phase-separation will increase or not with the 915 nm NIR light propagating through it, THFOW was heated by using a heat source, and the temperature of THFOW was recorded by an IR camera. Supplementary Fig. 12 shows that after the temperature of the THFOW raised to LCST (70 °C), the temperature was decreased with the heat source removed immediately, indicating that the THFOW after phase-separation would not absorb more laser energy, this is because the NIR laser light was scattering to the surrounding tissue.

## Soft tissue-affinity

To implant the THFOW in vivo, stability and biocompatibility with the organism are very important[16,26]. Fig. 5a shows the swelling behavior of the fabricated THFOWs in water to simulate internal wet environment of a living body. The results show that all the samples were stable in the water environment after swelling for 1000 min. Interestingly, a phenomenon that the sheath Na-alginate hydrogel could limit the swelling of the core ($N_{70}D_{30}$)$_{40}$ hydrogel was observed here. This was proved when by the core layer hydrogel, was swollen 110–120% after removing the sheath layer (Supplementary Fig. 13, and an additional explanation was in Supplementary Information). This is because the swollen core ($N_{70}D_{30}$)$_{40}$ hydrogel would pinch outward, causing shrinkage of the Ca-alginate layer hydrogel, and this bound sheath layer could withstand the swelling of the core layer. Thus, constructing a sheath hydrogel layer with a small swelling ratio that can restrict the swelling of the core hydrogel layer with a high swelling ratio is an efficient method. Such anti-swelling behavior could prevent the THFOW from swelling-induced degradation, and ensure its steady long-term use[27].

The mechanical properties of THFOWs are one of the important performance factors for its long-term use in vivo. Usually, a large differentiation of modulus between the human tissue and an implanted material would cause damage or inflammation to the host tissue, posing a threat to human health[28,29]. As the diameter of THFOW reduces, its tensile strength reduces. For the THFOW diameters ranging from 2417 μm/2533 μm to 633 μm/781 μm, their tensile strength reduces from 485.2 kPa to 246.9 kPa, and Young's modulus from 315.5 kPa to 149.1 kPa, respectively. (Fig. 5b and Supplementary Fig. 14). The Young's modulus was approximately five orders of magnitude less than that of silica-based optical fiber ($10^4$–$10^5$ MPa), which is much smaller than that of the polymer-based optical fibers ($10^3$ MPa). The

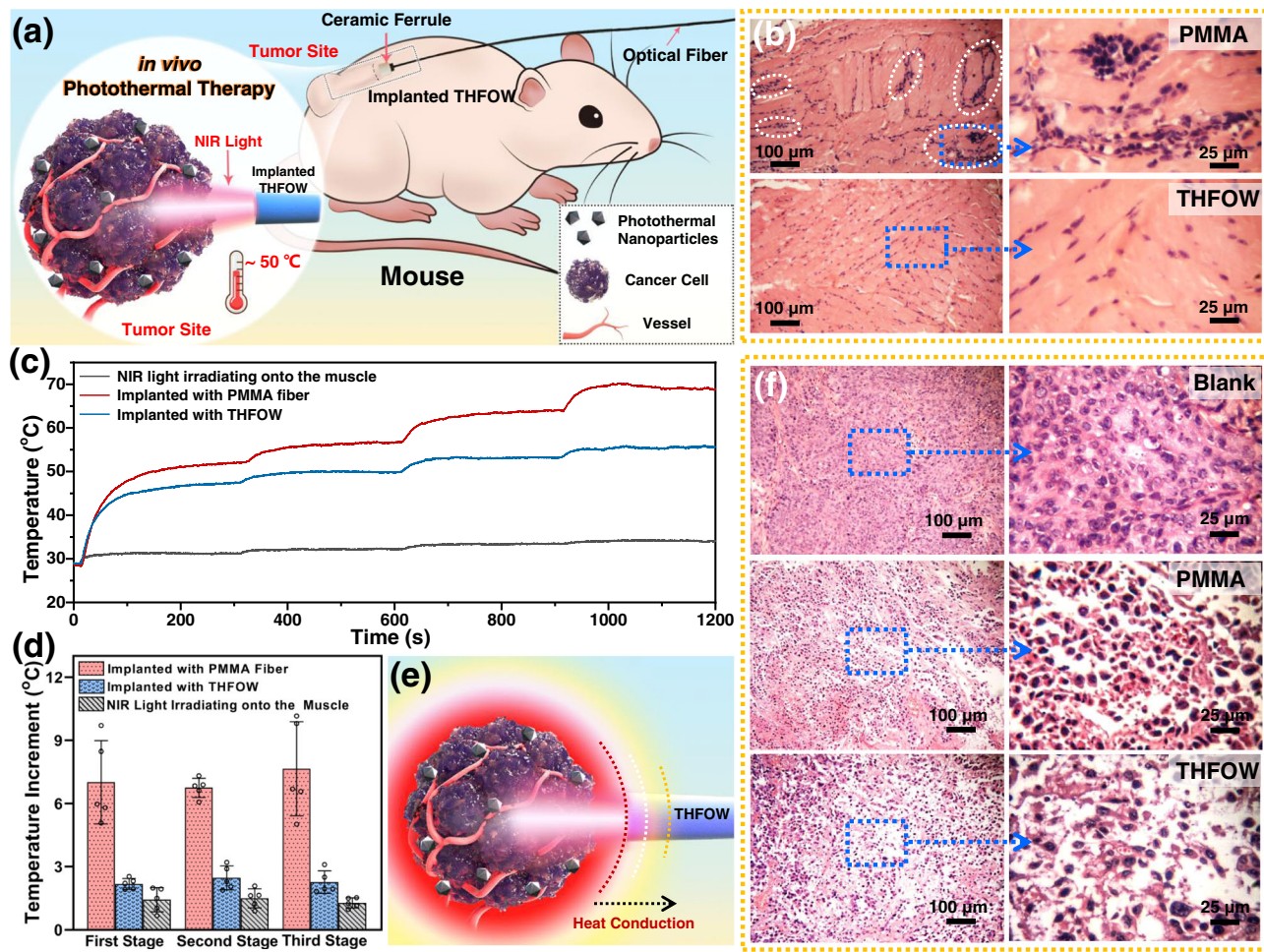

**Fig. 6 | Controllable photothermal therapy applied by the THFOW. a** Schematic illustration of the controllable photothermal therapy in vivo; (**b**) H&E stained muscle slices harvested from the tissues of mice in contact with PMMA fiber and THFOW; (**c**) Temperature changes of tumors/muscle monitored by the infrared thermal camera during laser irradiation; (**d**) Temperature increment during the photothermal therapy of the experimental and control groups ($n = 5$ biologically independent mice), data were presented as mean ± SD; (**e**) Schematic illustration of the heat conduction on the tumor site; (**f**) H&E stained tumor slices harvested from the blank, experimental and control groups after photothermal therapy. A representative image of three independent samples from each group is shown in **b** and **f**.

difference between THFOW and polymer-based optical fiber could be visually confirmed from Supplementary Fig. 15: The PMMA fiber remained unchanged state even after hanging a weight of 500 g to it; on the contrary, THFOW was easily deformed. The Young's modulus of THFOW was similar to that of the other hydrogel-based optical fibers (Fig. 5c), such as glucose-sensitive hydrogel optical fibers (-10 MPa) and strain sensor optical fibers (-200 kPa). Moreover, Young's modulus of the THFOW is similar to that of soft tissues in human body, such as brain tissue (0.005–0.06 MPa), liver (0.05–0.25 MPa), skin (0.8–40 MPa), and cartilage (30–100 MPa), ensuring high application potential for delivering light signals or energy within the human body. Here, we chosen porcine tissue as the model to simulate the light propagation under the tissue. Figure 5d shows the result of light propagation through THFOW in the porcine tissue. It can be found that light can propagate efficiently through THFOW, and the measured scattered light loss of profile is 0.65 dB cm$^{-1}$, which is similar to some of other reported works[6,18,30].

Cytotoxicity is the most important parameter for implanting the THFOW into tissues. Here, cell viability and live/dead assays of HeLa cells on THFOW were performed. The results show that only a slightly lower absorbance value (Supplementary Fig. 16a) and cell viability (Supplementary Fig. 16b) were observed for THFOW compared to a blank sample, which indicates that THFOW did not suppress cell growth around itself. In addition, this low cytotoxicity also could be proved by the results of the live/dead assay (Fig. 5d), it was found that the conditions for the experimental group and blank group were almost the same. Thus, THFOW possesses remarkable soft tissue-affinity and can be used for transporting light signals or energy in vivo.

## Thermal regulated interventional photothermal therapy

Based on the excellent light-guiding property, high thermal sensitivity, and remarkable soft tissue-affinity of the fabricated THFOW, we further explored the thermal regulated interventional photothermal therapy by delivering NIR laser light through it, as illustrated in Fig. 6a. As for animal models, mice implanted with THFOW were used as the experimental group, mice implanted with PMMA fiber were used as the control group, and mice implanted with nothing were used as the blank group. Histological evaluation of the mouse implanted with the PMMA fiber (Fig. 6b top part) shows that there were major immune-cell infiltrations, which were in contact with the PMMA fiber. In contrast, histological evaluation of the mouse implanted with THFOW were no major immune-cell infiltrations (Fig. 6b, lower half). This indicates good biocompatibility between THFOW and tissues.

In order to evaluate the thermal regulated interventional photomedicine by THFOW, a three-step increase of in vivo photothermal heating is performed. Firstly, all the 915 nm NIR light output intensity at the end of the PMMA fiber and THFOW were controlled at 210 mW, Fig. 6c showed that both the experimental and control groups had the

same heating rate during the first 1 min. When the temperature reached ~40 °C, a much lower heating rate was found in the experimental group, which was due to phase separation occurring in the THFOW. To further prove the effectiveness of temperature controllability in the experimental group, at the therapy time of 5, 10, and 15 min, we increased the NIR intensity by 5%, respectively. Results (Fig. 6d) show that temperature rises of $2.18 \pm 0.23$, $2.47 \pm 0.50$ and $2.26 \pm 0.48$ °C were founded at every stage for the group of mice implanted with THFOW. Which are much lower than the temperature rises in the group of mice implanted with PMMA. Due to the heat conduction in the tumor site (Fig. 6e), only a few parts of THFOW that contact with the tumor would undergo phase-separation during the three-step increase of in vivo photothermal heating. In this case, the temperature would stay at an equilibrium that transported NIR light to keep the thermal heating. Finally, by using the THFOW as the light-guide, the temperature could be efficiently controlled at 55.6 °C after the three-step increase of in vivo photothermal heating, where the PMMA fiber used was 69.2 °C (Fig. 6c and Supplementary Fig. 17). These results show that the THFOW could not only propagate light energy to the fixed site in vivo, but could also intelligently control the temperature during photothermal therapy. Thus, this intelligent response of THFOW could ensure efficient elimination of tumor cells, while lowering the risks of the overtemperature that causes the death of normal cells around the tumor site.

After several therapy sessions, tumors in the experimental and control groups were smaller and darker red than those in the blank group (Supplementary Fig. 18). The tumor Fig. 6f shows the H&E stained tumor slices harvested from the blank, experimental and control groups after photothermal therapy. It can be found that the histological photographs of the blank group exhibit normal cell morphologies. In contrast, the histological photographs of PMMA and THFOW can find the damaged nucleus and plasma membrane, confirming that cancer cells in vivo can be efficiently killed by the photothermal effects. However, at the same time, high temperature also would kill the healthy cells that can damage normal tissues adjacent to the lesions due to massive heat transfer, therefore, leading to more side effects and inhibiting the therapeutic accuracy of photothermal therapy. In this work, to compare the normal tissue damage between the controlled and uncontrolled photothermal therapy, tissues under the tumor site were harvested from the mice after photothermal therapy for evaluation, Supplementary Fig. 19 shows that among the same position of tissue, the PMMA group's slice showed the largest extent of damage, indicating that higher temperature would cause more serious damage to the healthy cells, and controlled temperature during the photothermal therapy can lower the risk of the damage of normal tissue. Finally, the implanted THFOW was taken out for evaluating the in vivo degradation, Supplementary Fig. 20 shows that after 1-week of implantation into the mouse, sheath calcium alginate hydrogel fiber became rougher compared that before implantation, which is due to the partial degradation in vivo, and further transported light test shows that light intensity transported have still remained $93.2\% \pm 5.4\%$.

In summary, a temperature-adaptive hydrogel fiber-based optical waveguide was fabricated through an integrated homogeneous-dynamic-crosslinking-spinning method and used for intelligent interventional photomedicine. The low temperature bath could ensure the polymerization happening within a single-phase, thus leading to the homogeneous formation of polymer networks, guarantee the high transparence of the THFOW. As a result, the fabricated THFOW possesses an excellent mechanical property (Young's modulus of 315.5–149.1 kPa), and optical-propagation property (−0.32 dB cm$^{-1}$ with 915 nm laser light). Furthermore, the core-sheath structure of the THFOW could limit the swelling behavior of the core functional gel. In addition, the fabricated THFOW is applied to demonstrating a thermal regulated interventional photomedicine, which was proposed here to

efficiently eliminate the tumor cells, meanwhile lower the risks of overtemperature (may seriously damage normal cell death around the tumor site). With remarkable soft tissue affinity, good thermo-sensitivity and excellent light propagation, the THFOW is promising to be broadly applied in the field of intelligent photomedicine.

## Methods

**Materials.** N-isopropylacrylamide (NIPAM), Poly(ethylene glycol) diacrylate (PEGDA, $M_n = 700$ Da) and photoinitiator 2-hydroxy-4′-(2-hydroxyethoxy)−2-methylpropiophenone (IRGACURE 2959, I2959, 98%) were bought from Sigma-Aldrich; N,N-Dimethylacrylamide (DMAAm, $M_n = 99$ g mol$^{-1}$, 99.0%) was bought from Tokyo Chemical Industry Co., Ltd (TCI); Sodium alginate (Na-alginate) was purchased from Sinopharm Chemical Reagent Co., Ltd.; Calcium chloride (CaCl$_2$), was supplied by Shanghai Lingfeng Chemical Reagent Co., Ltd; Deionized water was obtained from a water purification system (Heal Force Bio-Meditech Holdings Ltd.).

**Investigation and selection of N-isopropylacrylamide/N, N-Dimethylacrylamide based thermal hydrogel.** NIPAM, DMAAm, PEGDA and I2959 were added into the deionized water under continually magnetic stirring in dark to prepare the pre-gel solution, monomer (NIPAM + DMAAm) concentration was kept at 50 wt.%. PEGDA was used as a crosslinker, which the weight ratio of NIPAM + DMAAm to PEGDA was 100:1. I2959 was used as the photo-initiator and the ratio of I2959 to monomers = 1000:5. And the variation of the ratio between NIPAM and DMAAm (100:0, 90:10, 80:20, 70:30, 60:40 and 50:50) was used to investigate the thermo-sensitivity and transmittance of N-isopropylacrylamide/N, N-Dimethylacrylamide based thermal hydrogel. All the pre-gel solution was initiated by a mercury lamp UV irradiation (S1500, EXFO, Canada. $\lambda = 360$ nm, 2.77 W cm$^{-2}$), and irradiated under the UV light for 15 min. The detailed pre-gel solutions were as shown in Supplementary Table 1, and the samples were defined as (N$_X$D$_{100-X}$)$_{50}$, where the X = mass wt.% of NIPAM in the monomers.

**Preparations of temperature-adaptive hydrogel fiber-based optical waveguide (THFOW).** The THFOWs were fabricated by using integrated dynamic wet spinning. In brief, a 2.0 wt.% Na-alginate solution was chosen as the sheath spinning solution, while (N$_{70}$D$_{30}$)$_{50}$ solution was chosen as the core spinning solution. Firstly, these two solutions were injected into the coagulating solution (CaCl$_2$) through a coaxial needle by using two metering pumps (KDS100, KD Scientific, USA), where coaxial needle used to have four different types: the size of 2.00/1.43, 1.47/0.79, 1.17/0.71, 0.72/0.45 (mm/mm, sheath diameter/core diameter, as seen in Supplementary Fig. 3). The extrusion rate ratio between the core and sheath was kept at 1, where the extrusion rate of the both layer is 5 cm min$^{-1}$, and the drawing speed was kept at 5 cm min$^{-1}$. Then, a mercury lamp UV irradiation (S1500, EXFO, Canada. $\lambda = 360$ nm, 2.77 W cm$^{-2}$) was placed in the water bath to initiate the polymerization of the extruded core monomer solution. After that, the fabricated core-sheath hydrogel fiber was collected on a roller outside the water bath. Finally, the THFOW was immersed in the deionized water for 3 days to remove the unreacted monomers. The schematic illustration of fabricating process was shown in Fig. 2a.

**Characterizations of (N$_X$D$_{100-X}$)$_{50}$ thermal hydrogel and THFOW.** The optical transmittances (the wavelength range is 400–1100 nm) of (N$_X$D$_{100-X}$)$_{50}$ thermal hydrogels (gelled in a highly transparent polymer cube, size of $1 \times 1 \times 3$, length × width × high) were carried out by a TU1901 UV-vis spectrometer (Purkinje General). The temperature-dependent transmittances of (N$_X$D$_{100-X}$)$_{50}$ thermal hydrogels were also carried out by the TU1901 UV-vis spectrometer (Purkinje General) at the wavelength of 515 nm from the 20–70 °C, which equipped with a thermoregulator (±0.1 °C), and the heating rate was 0.5 °C min$^{-1}$. The lower critical solution temperature (LCST) of the hydrogel was determined at the middle point of maximal transmittance. The refractive index of the Ca-alginate and (N$_X$D$_{100-X}$)$_{50}$ hydrogels were carried out by an Abbe refractometer (Shanghai CSOIF Co., Ltd.). Photos of THFOW,

and laser light propagated through the THFOW were taken by a digital camera (Canon, EOS 80D), which the photo parameters are 1/1600 of speed, 1600 of ISO, and F3.5. Diameters of the THFOW were measured from the micrographs of THFOW taken by a microscopic camera (H1605, CCD camera, Shanghai Renyue Electronic Technology Co., Ltd.). And sheath thickness of the THFOWs was carried out from the photos taken by stereoscopic microscopy (SMZ745T, Nikon, Japan), core diameter was calculated by subtracting the diameter of THFOW to sheath thickness. The cross-sectional morphologies of the fabricated THFOWs were carried out by a field emission scanning electron microscope (FE-SEM, SU8010, Hitachi, Japan). The photos of THFOW under different temperature condition were taken by the digital camera (Canon, EOS 80D) using a lens (Canon Macro EF 100 mm 1:2.8 L IS USM), and the light intensity through the THFOW under temperature variation was measured by a handheld optical power meter (Newport, 1919R), during the heating-cooling recycle process, the temperature of the heating source about 270 °C, and the cooling process was driving by the room-temperature (22.5 °C). The $^{1}$H nuclear magnetic resonance (NMR) spectra under different temperature (31 °C–61 °C) were carried out by an NMR spectrometer (Bruker Avance 400) using $D_2O$ as a solvent with an increment of 3 °C. The phase-separated fraction ($p$) was used to characterize the degree of phase transition, and calculated by the equation of $p = (I_0 - I)/I_0$, where I and $I_0$ represent the normalized integrated intensities of a selected resonant peak at a specified temperature and 31 °C.

Light propagation test was carried out by focusing the different laser light ($\lambda$ = 450 nm, 515 nm and 650 nm) on one fiber tip and measuring the scattered light intensity over the hydrogel fiber lengths (15 cm), and the light propagation profile was analyzed by using the software Image J (Version of 1.51j8), and the photos were taken by the digital camera (Canon, EOS 80D) using a lens (Canon Macro EF 100 mm 1:2.8 L IS USM). Light propagation property of the THFOW with 915 nm NIR laser was carried out by using the method of cutback technique with an interval of 1 cm. Light intensity propagation during cycling heating and cooling process through the THFOW was measured by a handheld optical power meter (Newport, 1919R). Temperature of the THFOW that coupling with 915 nm NIR laser light (200 mW was adjusted at another tip of THFOW) is recorded by the IR camera before/after phase-separation. The mechanical properties of the THFOW were performed on an Instron 5969 universal test machine: 10 mm of the gauge length, and 20 mm min$^{-1}$ of the extension rate were used for the test, five of each fiber were tested to calculate the mechanical performance, such as strength, modulus, and elongation. Swelling behaviors of the THFOW was investigated by immersing the THFOW in water, and weighting at a predefined time. The swelling ratio of the THFOWs was carried out by $W_X/W_0$, where $W_X$ is the mass value of THFOW immersed in water for X min, and the $W_0$ represents the mass value of THFOW firstly fabricated. A further swelling of THFOW with no sheath layer was performed, and after swelling to the equilibrium state, the micrograph of the core only fiber was taken by the microscopic camera (H1605, CCD camera, Shanghai Renyue Electronic Technology Co., Ltd.), and the diameters were calculated from the micrograph.

Cell viability assay. Hela cells were used as the model cell. Firstly, Hela cells were seeded on 96-well plates (cell density was controlled at 10,000 cells per well), and adhered to the plates for 12 h before the assays. The cells were co-incubated with/without THFOW for 1–3 days, which were set as THFOW group and blank group (three samples were tested for every group), respectively. Then, 10 uL of CCK8 was added to each well, and incubated for another 2 h at 37 °C in the dark. Finally, by using a microplate reader (Bio-Rad, USA), the absorbance of CCK8 at 450 nm was measured as the absorbance value and to determine the cell viability.

Live/dead viability assay. Hela cells were obtained from American type culture collection. Hela cells were seeded on 24 well plates (cell density of 10,000 cells per well and allowed to adhere for 12 h before

the assays). The cells were co-incubated with/without THFOW for 1–3 days, which were set as THFOW group and blank group (three samples were tested for every group). After incubation, the spent media was aspirated out, and the cells were rinsed twice with PBS. Subsequently, Calcein-AM was first added into the well and the plates were incubated at 4 °C for 15 min. And washed with PBS, then Propidium iodide (PI) was added to dye for 5 min. Finally, a fluorescence microscope (LEICA DMi8, Germany) was used to record the resultants.

Thermal regulated interventional photothermal therapy. Five-week-old BALB/c nu/nu mice (19 ± 2 g) were used as animal model in this work, which purchased from Shanghai Slac laboratory animal Co. Ltd, and all animal experiments were carried out with the permission of the Ethical Committee for Animal Care of Donghua University. Mice were housed in groups of 5 mice per individual cage in a 12-h light-dark cycle (8:00–20:00 light; 20:00–8:00 dark), where the environment is kept at constant room temperature (21 ± 1 °C) and relative humidity of 40–70%. In detail, the THFOW$_{2500}$ (length of 2.0 cm) was firstly coupled with a ceramic ferrule with an inner diameter of 2.5 mm. Then, the THFOW was implanted into the backside of the mouse, and the uncoupled tip of THFOW$_{2500}$ was targeted to the tumor site (intratumorally inject with $Bi_2WO_{6-x}$ nanoparticles as a photothermal reagent), another tip was retained outside the skin, as seen in Fig. 6a and Supplementary Fig. 18. PMMA fiber was treated in the same process and used as the control group. To begin the photothermal therapy, the extracorporeal THFOW was coupled with an optical fiber to deliver the laser light from the light source. The temperature change of the mice during in vivo photothermal therapy was recorded by the photothermal therapy-monitoring system FLIR-A300 (FLIR Systems Inc., USA). After the in vivo photothermal treatment (the therapeutic period is 1-week), mice were sacrificed and the tumors were taken out and fixed in 10% formaldehyde solution for the pathological section. The specimens were imaged by an optical microscope (VHM2600, VIHENT, China). After finishing the therapy, the implanted THFOWs were taken out, surface morphology was observed by the digital camera and a microscopic camera (H1605, CCD camera, Shanghai Renyue Electronic Technology Co., Ltd.), and transported light intensity was detected by using the handheld optical power meter (Newport, 1919R).

Software. All the statistical analyses were carried out by Origin 8.5, Excel 2016 and GraphPad Prism 6. Chemical structures in the Figs. 2 and 4 were performed on ChemDraw Ultra 12.0. Images in Figs. 1, 2a, b, 4f, 6a, e, and Supplementary Fig. 19a are drawn by the soft of Blender (Version of 3.3.1 LTS) and PowerPoint.

### Reporting summary
Further information on research design is available in the Nature Portfolio Reporting Summary linked to this article.

## Data availability
All the data supporting the work are available within the main text of this article and its Supplementary Information. The raw data of Figs. 3–6, and Supplementary Figs. 1, 2, 4, 7, 11–14, 16 can be found in the Source Data. Source data are provided with this paper.

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

## Acknowledgements

This work is supported by National Key Research and Development Program of China (2021YFA1201302/2021YFA1201300 M.Z.); the National Natural Science Foundation of China (NO. 52173029 K.H.; NO. 51733002 M.Z.; NO. 51803022 K.H.); the Science and Technology Commission of Shanghai Municipality (No. 20JC1414900, M.Z.); the Natural Science Foundation of Shanghai (21ZR1400500 K.H.); Guoyin Chen thanks the support from the fellowship of China National Postdoctoral Program for Innovative Talents under Grant BX20220063 G.C., and Graduate Student Innovation Fund of Donghua University (CUSF-DH-D-2020038 G.C.).

## Author contributions

M.Z., K.H., and R.C. conceived the idea and supervised the research. G.C. conducted the experiments and analyzed the data. N.Y., P.W., T.C. and H.L. performed the experimental section of the soft tissue-affinity, and analyzed the data. C.Z., S.W., L.Z., and B.H. contributed data analysis and paper revision. All authors discussed the results.

## Competing interests

The authors declare no competing interests.
