## [Peer Review File · Nature Communications]

Temperature-Adaptive Hydrogel Optical Waveguide with Soft Tissue-Affinity for Thermal Regulated Interventional PhotomedicineREVIEWER COMMENTS

Reviewer #1 (Remarks to the Author):

This manuscript reports a thermal regulated interventional therapy based on a temperature-sensitive hydrogel optical fiber. This unique fiber shows high transmittance (0.17 dB cm⁻¹ at 650 nm) when the temperature is than the lower critical solution (LCST), but can become almost opacity when the temperature is slightly higher. Therefore, it can automatically cut off the light during the therapy when overheating, and lower the risk of overtemperature and the damage to healthy cells around the tumor. Besides, the authors develop an efficient dynamic method to fabricate the fiber. The obtained fibers are uniform and show a good soft tissue-affinity, low cytotoxicity, swelling stability. However, I do have a few important questions and comments about evidences presented by the authors. The authors should address them to make this work suitable for publication in highly selective Nature Communication journal.

Major comments

(1) In Figure 2, the light transmission properties of the hydrogel were carefully evaluated at three different wavelengths (450, 515, 650 nm) by the authors (Figure 2), however none at the actual working wavelength (915 nm) of the fiber. The authors should provide this essential data, and conduct the experiments in Figure 3a, b, c using this wavelength again.

(2) The fundamental working principle of this work relies heavily on the thermal properties of the hydrogel material, such as thermal conductivities and light-to-thermal conversion rates at both lower and higher of the LCST, heat capacity, etc. This information is essential to understand the whole process. For example, when temperature is higher than the LCST, the fiber will become opacity and absorbed more the laser energy. Thus, in this case, if the heat dissipation to the tissue is slow than light-to-thermal conversion at the LCST, the fiber will become even hotter and remain block forever. Thus, the fiber will never work.

(3) In my view, the evidences presented in Figure 5 are from convincing. (i) The authors used an infrared (IR) camera to evaluate the resulting thermal therapy performances in vivo. It is widely known that for IR camera is not accurate (error in a few degrees) in measuring temperature. Therefore, the observed the difference in Figure 5c, d could be due to the biological individual differences (such as the thickness of the skin fat). I suggested the authors use an implantable temperature sensor instead. (ii) More likely, the observed the difference could also due to the unevenly distribution of Bi₂WO₆x nanoparticles (photothermal reagent) on both tumor sites. (iii) Judging from Figure 5c, the light dispersion angles of THFOW and PMMA is not same. Therefore, even from the the same distance to the tumor sites, even assuming the distribution of the nanoparticles is the same, the resulting heating effects are not the same. I suggest the authors design a more conclusive experiment instead.

(4) The authors chose 48 °C as the temperature threshold. Is there any evidence that at this temperature could effectively kill the tumor? Is there any evidence suggested that a few degree higher (52 °C) could cause serious damage to healthy cells?

(5) It is really to say that the results presented in Figure 5e and Figure S13 can prove the THFOW fiber could efficiently eliminate cancer cells and lower normal cell. Is there any quantitative data could support this claim? And, the author should perform more repeated measures, in order to eliminate the individual biological differences.

(6) What is the long-term stability of the fiber after implantation? any influences to the light transmission property?

(7) In figure 5C, after the increasing the NIR intensity and the fiber was cut off, can author observe any temperature rise in the THFOW fiber?

Minor comments:

(1) The Figures of this manuscript should be reorganized. In my opinion, some of the data are not so important and should be put in the Supporting information instead, such as Figure 1g, Figure 3e, Figure 4d, 4e.

(2) The abbreviations in Figure 4a (THFOW-1, etc) are confusing. Please use the actual values. And why the diameters of fiber are so important? Too much effort.

(3) In Figure 3c, the exact heating and cooling points in each cycle are not clear? What are the exact temperatures of heating and cooling?

(4) The "blank" in Figure 5D means?

(5) the temperature rises in Line 302 are obtained by?

Reviewer #2 (Remarks to the Author):

The authors present temperature-gated hydrogel optical fiber for the treatment of tumor tissue in vivo. The core of the optical fiber was made of temperature-responsive hydrogel, pNIPAM, and PEGDA, which can change its transparency depending on the surrounding temperature. Optical performance, mechanical property, and biocompatibility of the fiber were characterized and in vivo demonstration for the photo-thermal treatment of the tumor tissue was performed.

The concept of this work is interesting and significant for medical applications. The key novel idea of this work is spontaneous on/off switching of optical fiber to avoid the overheating of healthy tissue during the photo-thermal treatment of the tumor. The basic performance of this temperature-triggered optical switching of the proposed hydrogel optical fiber was shown, however, the results are insufficient to justify their claims. Especially, the hydrogel optical performance on NIR laser is needed for proving the actual effectiveness of their system. Thus the referee thinks this paper does not reach the standard for publication in Nature Communications. Detailed comments are as follows:

1. The authors examined the light propagation characteristics of the THFOWs in Figure 2, but all of the data were obtained in the visible light wavelengths (450-650 nm). In addition to them, optical performance for the NIR region (915 nm) must be characterized in terms of light attenuation.

2. Generally, the absorption of NIR to water is larger than that of visible light. In some conditions, NIR irradiation to water makes heat. The temperature rise may occur during NIR introduction to the THFOWs because the main component of the hydrogel is water. There may be a maximum NIR laser power limit for THFOWs. The authors should check this matter.

3. The in vivo stability of the calcium alginate sheath needs to be explained. Generally, the calcium alginate hydrogel will undergo degradation in vivo through the exchange with ions in the body fluid. After insertion into the mouse for the anticipated treatment time, the authors should examine whether the sheath of the alginate hydrogel remains stable.

4. In the in vivo demonstration, the referee understood that the core became opaque as the temperature of the tumor tissue rises. However, since the opaque core itself scatters NIR, further temperature elevation may be affected by the scattered light. The referee thinks that the authors need to show data that the temperature around the tumor tissue does not increase further beyond a certain point (\sim LCST) even if the NIR intensity is

increased by varying the intensity of the NIR. In the experiment in Figure 5, the intensity of the NIR was increased by 5%, but that is not enough for their claim because this was just a one-step increase. For example, it would be necessary to show the results of a multi-step increase in NIR intensity and the result that the increase in ambient temperature reaches a plateau.

5. In the in vivo experiment, the difference in the temperature rise of THFOW and PMMA optical fibers was explained by "(line 297) which was due to phase separation," but the LCST of the THFOW used was 48 degrees Celcius, as shown in Figure 3(a). The referee doesn't think phase separation occurs. It may be another reason. For example, the light attenuation of PMMA optical fiber is expected to be lower than that of THFOW, which may be the reason.

6. The photographs of the tissue sections in Figure 5(e) and Figure S13 may not be sufficient for the authors to conclude that "THFOW eliminated cancer cells more effectively than PMMA and reduced damage to surrounding normal cells" as they claim. Please provide detailed explanations and results, such as the location of the probe in relation to the tumor tissue, which region in the tumor tissue was imaged as the tumor section, and which location around the probe the normal cells are photographed.

Reviewer #3 (Remarks to the Author):

In this work, the core-sheath structure of the hydrogel fiber with temperature response was applied in the photothermal therapy. The core hydrogel fiber prepared in the ice-bath had homogeneous formation and high transparence, and the sheath hydrogel had lower refractive index, resulting in excellent light propagation property. The opacity changed with temperature dramatically gives promise in the temperature control of the interventional photothermal therapy. Overall, the manuscript reported a good design and proved the effective control of the photothermal therapy. However, there are a few questions that need to be answered before acceptance:

1. H&E stained tumor slices should be analyzed in detail.
2. The laser light used in the photothermal therapy is 915 nm, but there is few discussion about light attenuation of the THFOW in the 915 nm.
3. The wavenumber used in this paper is misleading, maybe the original meaning is wavelength.
4. The meaning and acquisition method of phase-separated fraction is missing.
5. In the method section, the drawing speed and extrude speed in the preparation of the THFOW is unclear, and the expression of LCST is incorrect.
6. In the supporting information, the explanatory text of figure s1, s2 does not match the figure, and the paragraph under figure s8 makes confusion.
7. Does the light attenuation of the THFOW change with the laser power density?
8. The model in Scheme 1 is shocking.

Response to Reviewers

To Reviewer #1:

Comment: This manuscript reports a thermal regulated interventional therapy based on a temperature-sensitive hydrogel optical fiber. This unique fiber shows high transmittance (0.17 dB cm^{-1} at 650 nm) when the temperature is than the lower critical solution (LCST), but can become almost opacity when the temperature is slightly higher. Therefore, it can automatically cut off the light during the therapy when overheating, and lower the risk of overtemperature and the damage to healthy cells around the tumor. Besides, the authors develop an efficient dynamic method to fabricate the fiber. The obtained fibers are uniform and show a good soft tissue-affinity, low cytotoxicity, swelling stability. However, I do have a few important questions and comments about evidences presented by the authors. The authors should address them to make this work suitable for publication in highly selective Nature Communication journal.

>>Response:

Thank you very much for the valuable comments on our manuscript. The manuscript has been modified carefully according to your suggestion, some additional information has been added to strengthen the submission, and changed parts were highlighted in red in the text.

Major comments:

Question 1: In Figure 2, the light transmission properties of the hydrogel were carefully evaluated at three different wavelengths (450, 515, 650 nm) by the authors (Figure 2), however none at the actual working wavelength (915 nm) of the fiber. The authors should provide this essential data, and conduct the experiments in Figure 3a, b, c using this wavelength again.

>>Response:

Thank you very much for the valuable comments on our manuscript. Since 915 nm NIR light is a kind of invisible light, which is hard to observe with naked eyes, and the digital camera is also insensitive to the NIR light, a photo taken by a digital camera is hard to observe the transmission state of the laser light as seen in Figure S16a, of which the transported light intensity is 210 mW (measured by the handheld optical power meter), but only a few transmitted NIR laser light is detected by the camera. So, to evaluate the light attenuation of 915 nm laser light through THFOW, we employ the method of cutback technique, which is different from the experiments in Figure 3a, b,

c, but is also an efficient method for the Optical fiber attenuation test (*Adv. Mater.*, **2016**, *28(46)*: 10244-10249; *Adv. Mater. Technol.*, **2020**, *5(12)*: 2000515). The result in Figure S12b shows that the optical loss increased exponentially (linearly in dB scale) with the fiber length (Figure S16b). The measured light loss of the THFOW in the air was -0.32 dB cm^{-1} at a wavelength (λ) of 915 nm. And the results were added to the revised manuscript on Page 11, lines 291 – 294, and highlighted in red.

Figure S16 (a) NIR light (915 nm) propagated through a THFOW₂₅₀₀; (b) NIR light loss in THFOW₂₅₀₀.

Question 2: The fundamental working principle of this work relies heavily on the thermal properties of the hydrogel material, such as thermal conductivities and light-to-thermal conversion rates at both lower and higher of the LCST, heat capacity, etc. This information is essential to understand the whole process. For example, when temperature is higher than the LCST, the fiber will become opacity and absorbed more the laser energy. Thus, in this case, if the heat dissipation to the tissue is slow than light-to-thermal conversion at the LCST, the fiber will become even hotter and remain block forever. Thus, the fiber will never work.

>>Response:

Thank you very much for the valuable comments on our manuscript. In this work, the temperature raising mainly relies on the photothermal effect of pre-injected photothermal nanoparticles under the irradiation of NIR light. And it is indeed that the working principle of this work will be affected by the thermal properties of the

hydrogel material. According to your suggestion, we have designed an experiment that can show how THFOW work after the temperature reaches the LCST. Firstly, 915 nm NIR light was coupled with THFOW, and the light intensity was controlled at 210 mW at another tip of THFOW, which the light intensity was used for in vivo photothermal therapy. In this case, THFOW was heated by using a heat source, after the temperature of the THFOW was raised to 70 °C, removed the heat source away. At the moment, a phase-separation was founded at the tip of THFOW, and the NIR laser light was scattered to the surroundings (as shown in Figure S11, middle-inserted photo). The temperature is decreased with the heat source removed immediately, indicating that the THFOW after phase-separation would not absorb more laser energy, this is because the NIR laser light was scattering to the surrounding tissue. As we know, the thermal conductivity of the air is 0.028 W/(m • k) (*J. physic. chem. ref. data*, **1985**, *14(1)*: 227-234), and the thermal conductivity of the tissue is in the range of 0.5 – 0.7 W/(m • k) (*Physiological measurement*, 2003, *24(3)*: 769), which is much smaller than air. Thus, we predict that heat dissipation to the tissue is much faster than the light-to-thermal conversion of the THFOW at the LCST. And the results have been added to the manuscript on pages 8 and 9, lines 228 – 235. Furthermore, an improved in vivo experiment is also carried out, and a three-step increase of in vivo photothermal heating is performed. Results in Figure 5c showed that during each temperature increase step, the temperature raising could be efficiently restrained by using THFOW, indicating that there is an efficient controlling of temperature by using THFOW.

Figure S11 State variation of THFOW before/after phase-separation (coupling with 915 nm NIR light).

Figure 5c Infrared thermal images of the mice in control and experimental groups during photothermal therapy.

Question 3: In my view, the evidences presented in Figure 5 are from convincing. (i) The authors used an infrared (IR) camera to evaluate the resulting thermal therapy performances in-vivo. It is wildly known that for IR camera is not accurate (error in a few degrees) in measuring temperature. Therefore, the observed the difference in Figure 5c, d could be due to the biological individual differences (such as the thickness of the skin fat). I suggested the authors use an implantable temperature sensor instead. (ii) More likely, the observed the difference could also due to the unevenly distribution of Bi₂WO₆x nanoparticles (photothermal reagent) on both tumor sites. (iii) Juggling from Figure 5c, the light dispersion angels of THFOW and PMMA is not same. Therefore, even from the same distance to the tumor sites, even assuming the distribution of the nanoparticles is the same, the resulting heating effects are not the same. I suggest the authors design a more conclusive experiment instead.

>>**Response:**

Thank you very much for the valuable comments on our manuscript. As you said, there is a difference between the in vivo body temperature and the surface temperature during photothermal therapy. Here, we have recorded the temperature of the skin on the top of the tumor and the temperature at the bottom of the tumor. The result (Figure R1) shows that before the photothermal heating, the temperature at the bottom of the tumor is higher than the temperature of the skin on the top of the tumor. After beginning the photothermal heating, there is a difference in the initial state of heating, and with the temperature raised to 45 °C, these two temperatures become gradually close, and the temperature at the bottom of the tumor is about 0.7 °C higher than the temperature upon the skin on the top of tumor. Furthermore, the IR camera is commonly used for detecting the temperature during photothermal therapy, in which the target is a subcutaneous tumor, such as Zhuang Liu's group from Soochow University (*Nano Lett.*, **2018**, 18(9):

6037-6044; *Adv. Mater.*, **2016**, 28(14): 2716-2723; *ACS Nano*, **2018**, 12(9): 9412-9422), Zhigang Chen' Group from Donghua University (*Nanoscale*, **2019**, 11(32):15326-15338; *Adv. Mater.*, **2011**, 23(31): 3542-3547). Considering the minor difference (0.7 °C) between the temperature of the skin and under the tumor, and to accurately observe the temperature raising of the tumor site without implanting a temperature probe, we choose the subcutaneous tumor instead of the tumor beneath the deep tissue. Thus, the IR camera could use for recording the temperature raising at the tumor site. Besides, the IR camera is a temperature detecting equipment, which can't detect the light, and the light dispersion angles of THFOW and PMMA that you mentioned is the heat distribution during the photothermal therapy. Now, since the in vivo experiment is re-performed, and all the implanted PMMA fiber or THFOW is placed end-to-tumor, as illustrated in Figure 5a, where the implants are placed contacted with the tumor, and the corresponding statement is also changed and added to the manuscript Pages 11 and 12, lines 302 – 318.

Figure R1 Difference of the temperature that recording by IR camera and implanted temperature probe.

In order to eliminate the effects of biological individual differences and the uneven distribution of $\text{Bi}_2\text{WO}_{6-x}$ nanoparticles, 5 mice for every group are performed for evaluating the thermal regulated interventional photomedicine. Firstly, all the 915 nm NIR light output intensity at the end of the PMMA fiber and THFOW were controlled at 210 mW, Figure 5c showed that both the experimental and control groups had the same heating rate during the first 1 min. When the temperature reached approximately 40 °C, a much lower heating rate was found in the experimental group, which was due to phase separation occurring in the THFOW. To further prove the effectiveness of temperature controllability in the experimental group, at the therapy time of 5, 10, and

15 min, we increased the NIR intensity by 5%, respectively. Results (Figure 5d) show that temperature rises of 2.18 ± 0.23 , 2.47 ± 0.50 and 2.26 ± 0.48 °C were founded at every stage for the group of mice implanted with THFOW. Which are much lower than the temperature rises in the group of mice implanted with PMMA. Due to the heat conduction in the tumor site (Figure 5e), only a few parts of THFOW that contact with the tumor would undergo phase-separation during the three-step increase of in vivo photothermal heating. In this case, the temperature would stay at an equilibrium that transported NIR light to keep the thermal heating. Finally, by using the THFOW as the light-guide, the temperature could be efficiently controlled at 55.6 °C after the three-step increase of in vivo photothermal heating, where the PMMA fiber used was 69.2 °C (Figure 5c and Figure S17). These results show that the THFOW could not only propagate light energy to the fixed site in vivo, but could also intelligently control the temperature during photothermal therapy. Thus, this intelligent response of THFOW could ensure the efficient elimination of tumor cells, while lowering the risks of the overtemperature that causes the death of normal cells around the tumor site.

Figure 5. Controllable photothermal therapy applied by the THFOW: (c) Temperature changes of tumors/muscle monitored by the infrared thermal camera during laser irradiation; (d) Temperature increment during the photothermal therapy of the experimental and control groups; (e) Schematic illustration of the heat conduction on the tumor site.

Question 4: The authors chose 48 °C as the temperature threshold. Is there any evidence that at this temperature could effectively kill the tumor? Is there any evidence suggested that a few degrees higher (52 °C) could cause serious damage to healthy cells?

>>Response:

Thank you very much for the valuable comments on our manuscript. It is reported

that when the temperature was higher than 42 °C, the cancer cell will be killed (*Nat. Commun.*, **2016**, 7(1):1-10; *Nat. Nano.*, **2011**, 6:28-32). Thus, in this work, when choosing the 48 °C as the temperature threshold, the cancer cell would be efficiently killed. And results in Figure 5f show the H&E stained tumor slices harvested from the blank, experimental and control groups after photothermal therapy. It can be found that the histological photographs of the blank group exhibit normal cell morphologies. In contrast, the histological photographs of PMMA and THFOW can find the damaged nucleus and plasma membrane, confirming that cancer cells in vivo can be efficiently killed by the photothermal effects. However, at the same time, high temperature also would kill the healthy cells that can damage normal tissues adjacent to the lesions due to massive heat transfer, therefore, leading to more side effects and inhibiting the therapeutic accuracy of photothermal therapy. In this work, to compare the normal tissue damage between the controlled and uncontrolled photothermal therapy, tissues under the tumor site were harvested from the mice after photothermal therapy for evaluation, Figure S19 shows that among the same position of tissue, the PMMA group's slice showed the largest extent of damage, indicating that higher temperature would cause more serious damage to the healthy cells, and controlled temperature during the photothermal therapy can lower the risk of the damage of normal tissue. The changed parts were as seen in the revised manuscript, Page 12, lines 324 – 338, and the Supporting Information.

Figure S16 (a) Schematic illustrate of the harvested normal tissue that for clinicopathologic analysis; (b-d) H&E stained muscle slices harvested from the tissues of mice under the tumor site for different groups.

Question 5: It is really to say that the results presented in Figure 5e and Figure S13 can prove the THFOW fiber could efficiently eliminate cancer cells and lower normal cell. Is there any quantitative data could support this claim? And, the author should perform more repeated measures, in order to eliminate the individual biological

differences.

>>**Response:**

Thank you very much for the valuable comments on our manuscript. According to your suggestion, to eliminate the effects of biological individual differences and the uneven distribution of $\text{Bi}_2\text{WO}_{6-x}$ nanoparticles, 5 mice for every group are performed for evaluating the thermal regulated interventional photomedicine. By using THFOW as the light-guide, temperature of the tumor site could be efficiently controlled at 55.6 °C after the three-step increase of in vivo photothermal heating, where the PMMA fiber used was 69.2 °C (Figure 5c). Furthermore, in order to compare the normal tissue damage between the controlled and uncontrolled photothermal therapy, tissues under the tumor site were harvested from the mice after photothermal therapy for evaluating, Figure S19 shows that among the same position of tissue, the PMMA group's slice showed the largest extent of damage (the diameter of damaged tissue is more than 700 μm), and the THFOW group's slice showed a small extent of tissue damage, which the diameter is about 300 μm . indicating that higher temperature would cause more serious damage to the healthy cells, and controlled temperature during the photothermal therapy can lower the risk of the damage of normal tissue.

Question 6: What is the long-term stability of the fiber after implantation? any influences to the light transmission property?

>>**Response:**

Thank you very much for the valuable comments on our manuscript. Thank you very much for the valuable comments on our manuscript. According to your suggestion, we test the THFOW after insertion into the mouse, and the results were shown in Figure S20. We found that after 1-week of implantation into the mouse, sheath calcium alginate hydrogel is partly degraded, and the light attenuation remained at $93.2\% \pm 5.4\%$. This is due to that within an optical fiber, the light transmittance mainly relies on the total reflection of light at the sheath/core interface, and the refractive index of the sheath hydrogel is smaller than that of core hydrogel (*Nat. Commun.*, 2021, 12(1): 1-12; *Adv. Mater.*, 2015, 27(27): 4081-4086). Thus, partial degradation of sheath Ca-alginate hydrogel still meets the conditions for light transmission, and only affect a few of the light transmission, which may be due to the irregular surface formed on the sheath hydrogel after degradation as seen in Figure S20. Furthermore, we have reported a hydrogel optical fiber that without a sheath layer, and have an excellent light

propagation property, in which the light attenuation is 0.26 dB/cm with 532 nm (*Chem. Mater.*, **2020**, 32(22):9675-9687), confirming that even without sheath hydrogel fiber, the air/hydrogel interface also meets the conditions for the total reflection of light. In this work, we mainly propose a new concept of using thermosensitive hydrogel to realize the controlled photothermal therapy of cancer in deep tissue, and the fabricated THFOW is satisfied for the application within the therapy period. And the additional information was added to the revised manuscript, page 12, lines 338 – 342. Further for long-term usage, we also constructed a hydrogel optical fiber that is resistant to degradation in vivo, which the sheath hydrogel fiber is constituted from double-network hydrogel. This hydrogel optical fiber is suitable for long-term in vivo photomedicine application and will be reported in the next research soon.

Figure S20 THFOW before/after implanted into the mouse for photothermal therapy.

Question 7: In figure 5C, after the increasing the NIR intensity and the fiber was cut off, can author observe any temperature rise in the THFOW fiber?

>>**Response:**

Thank you very much for the valuable comments on our manuscript. The temperature raising mainly relies on the photothermal effect of the photothermal nanoparticle under the NIR light irradiation, and after increasing the NIR intensity, the temperature raising is detected on the tumor site as shown in Figure 5c and Figure S14. In order to visually observe the thermal heating of THFOW under the THFOW, we have designed an experiment that can show how THFOW work after the temperature reaches the LCST as detailed described in the response to *Question 2*. In brief, Figure S11 shows that the THFOW after phase-separation would not absorb more laser energy,

indicating the heat dissipation to the tissue is much faster than the light-to-thermal conversion of the THFOW at the LCST. Thus, there is no temperature rise in the THFOW fiber after the fiber was phase-separation.

Figure S11 State variation of THFOW before/after phase-separation (coupling with 915 nm NIR light).

Minor comments:

Question 8: The Figures of this manuscript should be reorganized. In my opinion, some of the data are not so important and should be put in the Supporting information instead, such as Figure 1g, Figure 3e, Figure 4d, 4e.

>>Response:

Thank you very much for the valuable comments on our manuscript. The Figures in this manuscript were reorganized according to your kind suggestion. Figure 1g, Figure 3e, Figure 4d, and 4e were put into the Supporting Information as Figure S7. Figure S10, and Figure S15, respectively. And the changed part was highlighted in red in the manuscript and Supporting Information.

Question 9: The abbreviations in Figure 4a (THFOW-1, etc) are confusing. Please use the actual values. And why the diameters of fiber are so important? Too much effort.

>>Response:

Thank you very much for the valuable comments on our manuscript. According to your suggestion, we change the abbreviations of the THFOW with different diameters, such as THFOW₂₅₀₀, THFOW₁₅₀₀, THFOW₁₂₀₀, and THFOW₈₀₀, where the value represents the magnitude of the fiber diameter. Here we want to show the adjustability of diameter for THFOW, and give the promise of these THFOWs with different

diameters are suitable for various application scenarios, according to your suggestion, a part of the description about fiber size has been removed now. And the changed part in the manuscript was highlighted in red, page 5, line 149, Figure 1e, Figure 2c, Figure 4a, Figure S4 – S6, and Figure S13.

Question 10: In Figure 3c, the exact heating and cooling points in each cycle are not clear? What are the exact temperatures of heating and cooling?

>>**Response:**

Thank you very much for the valuable comments on our manuscript. We here use the mercury lamp UV irradiation (S1500, EXFO, Canada. $\lambda = 360$ nm, 2.77 W/cm²) as the heat source, where the temperature at the output of the irradiation is measured as about 270 °C. And the cooling process was driven by the room-temperature (22.5 °C). And this information was added to the manuscript, as seen in page 15, lines 424 – 425.

Question 11: The “blank” in Figure 5D means?

>>**Response:**

Thank you very much for the valuable comments on our manuscript. The “blank” in previous Figure 5d represents the temperature raising recording from the NIR laser light irradiating directly onto the mice’ muscle. And the expression has now been changed into “NIR light irradiating onto the muscle”, as seen in Figure 5c in the revised manuscript.

Question 12: The temperature rises in Line 302 are obtained by?

>>**Response:**

Thank you very much for the valuable comments on our manuscript. The temperature rises in line 302 were obtained from previous Figure 5d (now is Figure 5c), it refers to the increment of temperature at the tumor site after the light intensity increased.

To Reviewer#2:

Comment: The authors present temperature-gated hydrogel optical fiber for the treatment of tumor tissue in vivo. The core of the optical fiber was made of temperature-responsive hydrogel, pNIPAM, and PEGDA, which can change its transparency depending on the surrounding temperature. Optical performance, mechanical property, and biocompatibility of the fiber were characterized and in vivo demonstration for the photo-thermal treatment of the tumor tissue was performed.

The concept of this work is interesting and significant for medical applications. The key novel idea of this work is spontaneous on/off switching of optical fiber to avoid the overheating of healthy tissue during the photo-thermal treatment of the tumor. The basic performance of this temperature-triggered optical switching of the proposed hydrogel optical fiber was shown, however, the results are insufficient to justify their claims. Especially, the hydrogel optical performance on NIR laser is needed for proving the actual effectiveness of their system. Thus, the referee thinks this paper does not reach the standard for publication in Nature Communications. Detailed comments are as follows:

>>Response:

We thank reviewer#2 very much for the positive comments. The manuscript has been modified carefully according to your suggestion, some additional information have been added to strengthen the submission, and all the changed parts were highlighted in red in the text.

Question 1: The authors examined the light propagation characteristics of the THFOWs in Figure 2, but all of the data were obtained in the visible light wavelengths (450 – 650 nm). In addition to them, optical performance for the NIR region (915 nm) must be characterized in terms of light attenuation.

>>Response:

Thank you very much for the valuable comments on our manuscript. 915 nm laser light is a kind of invisible light, which is hard to observe with naked eyes, and the digital camera is also insensitive to the NIR light, thus the picture taken by a digital camera is hard to observe the transmission state of the laser light as seen in Figure S16a, of which the transported light intensity is 210 mW (measured by the handheld optical power meter), but only a few transmitted NIR laser light is detected by the camera. So, to evaluate the light attenuation of 915 nm laser light through THFOW, we employ the method of cutback technique, which is different from the experiments in Figure 3a, b, c, but is also an efficient method for the Optical fiber attenuation test (*Adv. Mater.*, **2016**, *28(46)*: 10244-10249; *Adv. Mater. Technol.*, **2020**, *5(12)*: 2000515). The result in Figure S12b shows that the optical loss increased exponentially (linearly in dB scale) with the fiber length (Figure S16b). The measured light loss of the THFOW in the air was -0.32 dB cm⁻¹ at a wavelength (λ) of 915 nm. And the results were added to the revised manuscript on Page 11, lines 291 – 294, and highlighted in red.

Figure S16 (a) NIR light (915 nm) propagated through a THFOW₂₅₀₀; (b) NIR light loss in THFOW₂₅₀₀.

Question 2: Generally, the absorption of NIR to water is larger than that of visible light. In some conditions, NIR irradiation to water makes heat. The temperature rise may occur during NIR introduction to the THFOWs because the main component of the hydrogel is water. There may be a maximum NIR laser power limit for THFOWs. The authors should check this matter.

>>Response:

Thank you very much for the valuable comments on our manuscript. According to your suggestion, we tested the maximum light intensity propagated through the TSOHF (length of 1.5 cm). An experimental device is set up for the testing as shown in Figure R2a, and two components were selected as observation targets, including the whole device and the hydrogel part (as shown in the photograph inserted in Figure R2b). As seen from the results (Figure R2b and Figure R2c), with the transported light intensity was increased from 200mW to 1100mW (detected by the handheld optical power meter), the temperature of the whole device and the hydrogel part were both increased, and the temperature of the coupling part is higher than the hydrogel part, which was due to that ceramic ferrule absorb the NIR light and generate more heat than hydrogel. Until the transported light intensity is up to 1100 mW, the temperature of the coupling part is higher than the LCST, while the hydrogel part is still lower than the LCST. Thus, there is a maximum NIR laser power limit for THFOWs, in which the transported light intensity is 1100 mW. Moreover, this light intensity corresponds to a light power of

20.37 W/cm², which is much larger than the light power used in usual photothermal therapy (usually ~1 W/cm²).

Figure R2 Testing of maximum light intensity: (a) Photograph of the testing device; (b) Temperature changes of the whole device and the hydrogel part during the test; (c) Photograph of the coupling part that the THFOW undergoing phase-transition.

Question 3: The in vivo stability of the calcium alginate sheath needs to be explained. Generally, the calcium alginate hydrogel will undergo degradation in vivo through the exchange with ions in the body fluid. After insertion into the mouse for the anticipated treatment time, the authors should examine whether the sheath of the alginate hydrogel remains stable.

>>**Response:**

Thank you very much for the valuable comments on our manuscript. Thank you very much for the valuable comments on our manuscript. According to your suggestion, we test the THFOW after insertion into the mouse, and the results were shown in Figure S20. We found that after 1-week of implantation into the mouse, sheath calcium alginate hydrogel is partly degraded, and the light attenuation remained at $93.2\% \pm 5.4\%$. This is due to that within an optical fiber, the light transmittance mainly relies on the total reflection of light at the sheath/core interface, and the refractive index of the sheath hydrogel is smaller than that of core hydrogel (*Nat. Commun.*, 2021, 12(1): 1-12; *Adv. Mater.*, 2015, 27(27): 4081-4086). Thus, partial degradation of sheath Ca-alginate hydrogel still meets the conditions for light transmission, and only affects a few of the light transmission, which may be due to the irregular surface formed on the sheath hydrogel after degradation as seen in Figure S20. Furthermore, we have reported a hydrogel optical fiber that without a sheath layer, and have an excellent light propagation property, in which the light attenuation is 0.26 dB/cm with 532 nm (*Chem.*

Mater., 2020, 32(22): 9675-9687), confirming that even without sheath hydrogel fiber, the air/hydrogel interface also meets the conditions for the total reflection of light. In this work, we mainly propose a new concept of using thermosensitive hydrogel to realize the controlled photothermal therapy of cancer in deep tissue, and the fabricated THFOW is satisfied for the application within the therapy period. And additional information was added to the revised manuscript, page 12, lines 338 – 342. Further for long-term usage, we also constructed a hydrogel optical fiber that is resistant to degradation in vivo, which the sheath hydrogel fiber is constituted from double-network hydrogel. This hydrogel optical fiber is suitable for long-term in vivo photomedicine application and will be reported in the next research soon.

Figure S20 THFOW before/after implanted into the mouse for photothermal therapy.

Question 4: In the in vivo demonstration, the referee understood that the core became opaque as the temperature of the tumor tissue rises. However, since the opaque core itself scatters NIR, further temperature elevation may be affected by the scattered light. The referee thinks that the authors need to show data that the temperature around the tumor tissue does not increase further beyond a certain point (\sim LCST) even if the NIR intensity is increased by varying the intensity of the NIR. In the experiment in Figure 5, the intensity of the NIR was increased by 5%, but that is not enough for their claim because this was just a one-step increase. For example, it would be necessary to show the results of a multi-step increase in NIR intensity and the result that the increase in

ambient temperature reaches a plateau.

>>Response:

Thank you very much for the valuable comments on our manuscript. In order to eliminate the effects of biological individual differences and the uneven distribution of $\text{Bi}_2\text{WO}_{6-x}$ nanoparticles, 5 mice for every group are performed for evaluating the thermal regulated interventional photomedicine. Firstly, all the 915 nm NIR light output intensity at the end of the PMMA fiber and THFOW were controlled at 210 mW, Figure 5c showed that both the experimental and control groups had the same heating rate during the first 1 min. When the temperature reached approximately 40 °C, a much lower heating rate was found in the experimental group, which was due to phase separation occurring in the THFOW. To further prove the effectiveness of temperature controllability in the experimental group, at the therapy time of 5, 10, and 15 min, we increased the NIR intensity by 5%, respectively. Results (Figure 5d) show that temperature rises of 2.18 ± 0.23 , 2.47 ± 0.50 and 2.26 ± 0.48 °C were founded at every stage for the group of mice implanted with THFOW. Which are much lower than the temperature rises in the group of mice implanted with PMMA. Due to the heat conduction in the tumor site (Figure 5e), only a few parts of THFOW that contact with the tumor would undergo phase-separation during the three-step increase of in vivo photothermal heating. In this case, the temperature would stay at an equilibrium that transported NIR light to keep the thermal heating. Finally, by using the THFOW as the light-guide, the temperature could be efficiently controlled at 55.6 °C after the three-step increase of in vivo photothermal heating, where the PMMA fiber used was 69.2 °C (Figure 5c and Figure S17). These results show that the THFOW could not only propagate light energy to the fixed site in vivo, but could also intelligently control the temperature during photothermal therapy. Thus, this intelligent response of THFOW could ensure the efficient elimination of tumor cells, while lowering the risks of the overtemperature that causes the death of normal cells around the tumor site.

Figure 5. Controllable photothermal therapy applied by the THFOW: (c) Temperature changes of tumors/muscle monitored by the infrared thermal camera during laser irradiation; (d) Temperature increment during the photothermal therapy of the experimental and control groups; (e) Schematic illustration of the heat conduction on the tumor site.

Question 5: In the in vivo experiment, the difference in the temperature rises of THFOW and PMMA optical fibers was explained by "(line 297) which was due to phase separation," but the LCST of the THFOW used was 48 degrees Celsius, as shown in Figure 3(a). The referee doesn't think phase separation occurs. It may be another reason. For example, the light attenuation of PMMA optical fiber is expected to be lower than that of THFOW, which may be the reason.

>>**Response:**

Thank you very much for the valuable comments on our manuscript. As for in vivo photothermal therapy in this work, all the 915 nm NIR light output intensity at the end of the PMMA fiber and THFOW were controlled at 210 mW, Figure 5c shows that both the experimental and control groups had the same heating rate during the first 1 min. When the temperature reached approximately 40 °C, a much lower heating rate was found in the experimental group, which was due to phase separation occurring in the THFOW. To further prove the effectiveness of temperature controllability in the experimental group, at the therapy time of 5, 10, and 15 min, we increased the NIR intensity by 5%, respectively. Results (Figure 5c) show that temperature rises of 2.18 ± 0.23 , 2.47 ± 0.50 and 2.26 ± 0.48 °C were founded at every stage for the group of mice implanted with THFOW. Which are much lower than the temperature rises in the group

of mice implanted with PMMA.

Question 6: The photographs of the tissue sections in Figure 5(e) and Figure S13 may not be sufficient for the authors to conclude that "THFOW eliminated cancer cells more effectively than PMMA and reduced damage to surrounding normal cells" as they claim. Please provide detailed explanations and results, such as the location of the probe in relation to the tumor tissue, which region in the tumor tissue was imaged as the tumor section, and which location around the probe the normal cells are photographed.

>>Response:

Thank you very much for the valuable comments on our manuscript. We are sorry for our expression caused by your misunderstanding. What we claim is not that "THFOW eliminated cancer cells more effectively than PMMA and reduced damage to surrounding normal cells", but is "this intelligent response of THFOW could ensure efficient elimination of tumor cells, while lowering the risks of the overtemperature that causes the death of normal cells around the tumor site". We didn't compare the therapeutic effect between using PMMA fiber and THFOW, and the main idea of this article is that the THFOW could efficiently eliminate the tumor cells while lower the risk of normal cell death surrounding the tumor site due to excessively high temperature.

Moreover, according to your advice, we have provided detailed explanations and results, as seen on Page 12, lines 324 – 339. That in this work, when choosing the 48 °C as the temperature threshold, the cancer cell would be efficiently killed. And results in Figure 5f show the H&E stained tumor slices harvested from the blank, experimental, and control groups after photothermal therapy. It can be found that the histological photographs of the blank group exhibit normal cell morphologies. In contrast, the histological photographs of PMMA and THFOW can find the damaged nucleus and plasma membrane, confirming that cancer cells in vivo can be efficiently killed by the photothermal effects. However, at the same time, high temperature also would kill the healthy cells that can damage normal tissues adjacent to the lesions due to massive heat transfer, therefore, leading to more side effects and inhibiting the therapeutic accuracy of photothermal therapy. In this work, to compare the normal tissue damage between the controlled and uncontrolled photothermal therapy, tissues under the tumor site were harvested from the mice after photothermal therapy for evaluating, Figure S19 shows that among the same position of tissue, the PMMA group's slice showed the largest extent of damage, indicating that higher temperature would cause more serious damage

to the healthy cells, and controlled temperature during the photothermal therapy can lower the risk of the damage of normal tissue.

Figure S19 (a) Schematic illustrate of the harvested normal tissue that for clinicopathologic analysis; (b-d) *H&E* stained muscle slices harvested from the tissues of mice under the tumor site for different groups.

To Reviewer #3:

Comment: In this work, the core-sheath structure of the hydrogel fiber with temperature response was applied in the photothermal therapy. The core hydrogel fiber prepared in the ice-bath had homogeneous formation and high transparency, and the sheath hydrogel had lower refractive index, resulting in excellent light propagation property. The opacity changed with temperature dramatically gives promise in the temperature control of the interventional photothermal therapy. Overall, the manuscript reported a good design and proved the effective control of the photothermal therapy. However, there are a few questions that need to be answered before acceptance.

>>Response:

Thank you very much for the valuable comments on our manuscript. The manuscript has been modified carefully according to your suggestion, some additional information have been added to strengthen the submission, and changed parts were highlighted in red in the text.

Question 1: H&E stained tumor slices should be analyzed in detail.

>>Response:

Thank you very much for the valuable comments on our manuscript. The detailed description of the H&E stained tumor slices is now added in the manuscript, Pages 11 – 12, lines 297 – 301, and lines 324 – 338.

Question 2: The laser light used in the photothermal therapy is 915 nm, but there are few discussions about light attenuation of the THFOW in the 915 nm.

>>**Response:**

Thank you very much for the valuable comments on our manuscript. 915 nm laser light is a kind of invisible light, which is hard to observe with naked eyes, and the digital camera is also insensitive to the NIR light, thus the picture taken by a digital camera is hard to observe the transmission state of the laser light as seen in Figure S16a, of which the transported light intensity is 210 mW (measured by the handheld optical power meter), but only a few transmitted NIR laser light is detected by the camera. So, to evaluate the light attenuation of 915 nm laser light through THFOW, we employ the method of cutback technique, which is different from the experiments in Figure 3a, b, c, but is also an efficient method for the Optical fiber attenuation test (*Adv. Mater.*, **2016**, *28(46)*: 10244-10249; *Adv. Mater. Technol.*, **2020**, *5(12)*: 2000515). The result in Figure S12b shows that the optical loss increased exponentially (linearly in dB scale) with the fiber length (Figure S16b). The measured light loss of the THFOW in the air was -0.32 dB cm^{-1} at a wavelength (λ) of 915 nm. And the results were added to the revised manuscript on Page 11, lines 291 – 294, and highlighted in red.

Figure S16 (a) NIR light (915 nm) propagated through a THFOW₂₅₀₀; (b) NIR light loss in THFOW₂₅₀₀.

Question 3: The wavenumber used in this paper is misleading, maybe the original meaning is wavelength.

>>**Response:**

Thank you very much for the valuable comments on our manuscript. Just as you said, the wavenumber was misused, and now has been corrected with “wavelength”, which was highlighted in red, on pages 6 – 7, lines 170, 179, and 180.

Question 4: The meaning and acquisition method of phase-separated fraction is missing.

>>Response:

Thank you very much for the valuable comments on our manuscript. According to your suggestion, the meaning and acquisition method of the phase-separated fraction is now added to the manuscript, and the changed part is highlighted in red as shown on Page 15, lines 409 – 410.

Question 5: In the method section, the drawing speed and extrude speed in the preparation of the THFOW is unclear, and the expression of LCST is incorrect.

>>Response:

Thank you very much for the valuable comments on our manuscript. The detailed speed of the drawing and extrude during the spinning is now given in the manuscript, pages 14 – 15, lines 394 – 396, and highlighted in red. And the expression of LCST in the method section is corrected in the manuscript, and highlighted in red (page15, line 409 – 410).

Question 6: In the supporting information, the explanatory text of figure s1, s2 does not match the figure, and the paragraph under Figure S8 makes confusion.

>>Response:

Thank you very much for the valuable comments on our manuscript. The figure caption of Figure S1 and Figure S2 was revised, and the changed parts were highlighted in red. The paragraph under Figure S8 was an additional explanation of the swelling behavior of the TSOHF after removing sheath hydrogel, and the related illustration was added in Supporting Information.

Question 7: Does the light attenuation of the THFOW change with the laser power density?

>>Response:

Thank you very much for the valuable comments on our manuscript. According to your suggestion, we have tested the light attenuation of the THFOW under different laser power. And the results in Figure S8 show that laser light with different power

could all propagate through the THFOW, and the attenuation of the light profiles shows that the power of the light didn't affect the attenuation of the light in THFOW. The additional part was highlighted in red in the manuscript (page 7, lines 183 – 186) and Supporting Information (Figure S8).

Figure S8 (a) Light transmission of THFOW₈₀₀ coupling with different laser intensities, scale bar is 1 cm; (b) Light loss through the THFOW₈₀₀ that coupling with different laser intensities.

Question 8: The model in Scheme 1 is shocking.

>>**Response:**

Thank you very much for the admire on Scheme 1 in our manuscript. In Scheme 1, we are aiming to descript a concept of thermal regulated interventional photomedicine accurately, and that is exactly what we are hoping for.

REVIEWER COMMENTS

Reviewer #1 (Remarks to the Author):

I have thoroughly checked the revised paper. The authors have addressed all my concern. I feel the paper could be considered for publication upon addressing some minor details: (1) In the Abstract, since the light attenuation at 650 nm is not important to the application, it is better to change to 915 nm.

(2) It is better to add the light attenuation at 915 to Figure 2c, which also means the aauthors should test the light attenuations for all diameters using the cut back method.

(3)the resolution of Figure S16 b-d now in the SI are far better than Figure 5f in the main text, why not change them?

(4) It is better to put result of light attenuation before and after implantation in Figure 4, it is far more important some the figure, such as the fiber diameters.

(5) Is 1-week implantation test enough long? How long does a photothermal therapy usually take?

Reviewer #2 (Remarks to the Author):

The authors have made satisfactory revisions to the manuscript and have addressed all the questions raised by the referee. The referee thinks this paper could meet the standard of Nature Communications.

Reviewer #3 (Remarks to the Author):

The authors have revised their manuscript according to the reviewers' comments. Now it is acceptable.

Response to Reviewers

To Reviewer #1:

Comment: I have thoroughly checked the revised paper. The authors have addressed all my concern. I feel the paper could be considered for publication upon addressing some minor details.

>>Response:

Thank you very much for the valuable comments on our manuscript. The manuscript has been modified carefully according to your suggestion, some additional information has been added to strengthen the submission, and changed parts were highlighted in red in the text.

Question 1: In the Abstract, since the light attenuation at 650 nm is not important to the application, it is better to change to 915 nm.

>>Response:

Thank you very much for the valuable comments on our manuscript. According to your suggestion, the description of the light attenuation in the abstract and other relative parts are now changed to 915 nm. The changed parts are highlighted in red in the manuscript, pages 1, 3, and 10; lines 19 – 21, 73, 327, respectively.

Question 2: It is better to add the light attenuation at 915 to Figure 2c, which also means the authors should test the light attenuations for all diameters using the cut back method.

>>Response:

Thank you very much for the valuable comments on our manuscript. According to your suggestion, we have tested the light attenuation of the 915 nm laser light by the cut back technique. Due to the different methods used, we have moved the result of light attenuation at 915 from Figure S16b to Figure 2d as shown in the manuscript, Page 18. And the corresponding description was added to a suitable position on Page 6, Lines 179-184, and highlighted in red.

Figure 2. Light propagation through the THFOW: (a) laser light with different wavelength ($\lambda = 450, 515$ and 650 nm) propagate through the THFOW; (b) Laser light propagates through THFOWs with different diameters; (c) Light attenuation of the THFOW calculated from the scattered light intensity along with the THFOW profile; (d) Propagation loss of the 915 nm laser light through THFOWs, measured by a cutback technique. Scale bars in (a), (b) are 1 cm.

Question 3: the resolution of Figure S19 b – d now in the SI are far better than Figure 5f in the main text, why not change them?

>>Response:

Thank you very much for the valuable comments on our manuscript. Maybe due to that in the process of image processing, the sharpness of the picture is damaged, causing a lower resolution of the photos. And according to your advice, we have now reproduced them, and Figure 5f is now changed a new version as seen on Page 20.

Question 4: It is better to put result of light attenuation before and after implantation in Figure 4, it is far more important some the figure, such as the fiber diameters.

>>Response:

Thank you very much for the valuable comments on our manuscript. According to your suggestion, we put the results of the light attenuation before and after implantation in Figure 4 as shown in the manuscript (page 19), and the fiber diameter is also added. The corresponding description was moved to page 8, lines 255 – 259, which was highlighted in red.

Figure 4. Compatibility of the THFOW with organisms: (a) Swelling behavior and (b) mechanical properties of the fabricated THFOWs; (c) Comparison of mechanical properties among the THFOW, tissues and other light-guide fibers; (d) Laser light ($\lambda = 515 \text{ nm}$) propagation within porcine tissue through implanted THFOW₂₅₀₀, scale bar = 1 cm. (e) Live/dead assay of HeLa cells on THFOWs compared to blank at 1 and 3 days, where live cells are in green, Scale bars: 100 μm .

Question 5: Is 1-week implantation test enough long? How long does a photothermal therapy usually take?

>>Response:

Thank you very much for the valuable comments on our manuscript. Due to that high temperature can effectively kill most of the cancer cells in the tumor site (generally more than 10 min of thermal heating). Thus, the therapeutic period of photothermal therapy can be arranged in several days. As far as we know, many of the reported photothermal therapeutic periods were usually around 10 days, for example, Meng et al reported a method of photothermal therapy that the therapeutic period was 7 days (ACS Nano 2018, 12, 9412-9422), 12 days of the therapeutic period was reported by Wang et al (Nanoscale 2019, 11, 15326-15338). In addition, due to the optical waveguide used, NIR laser could be efficiently delivered to the tumor site, which could enhance the precisely photothermal therapy for cancer therapy. Thus, we chose 7 days as the therapeutic period, and the results show that cancer cells are efficiently killed after treatment. Furthermore, the main idea of this article we delivered here is the new concept of “this intelligent response of THFOW could ensure efficient elimination of tumor cells while lowering the risks of the overtemperature that causes the death of

normal cells around the tumor site”. Which could efficiently eliminate the tumor cells while lower the risk of normal cell death surrounding the tumor site due to excessively high temperature.

REVIEWERS' COMMENTS

Reviewer #1 (Remarks to the Author):

The authors have made satisfactory revisions to the manuscript and have addressed all my concerns . Therefore, I have no further comments.